# Water quality assessment and source identification of the Shuangji River (China) using multivariate statistical methods

**Junzhao Liu**[1], **Dong Zhang** [2]*, **Qiuju Tang**[1], **Hongbin Xu**[1], **Shanheng Huang**[1], **Dan Shang**[1], **Ruxue Liu**[1]

**1** School of Ecology and Environment, Zhengzhou University, Zhengzhou, China, **2** School of Architecture, Zhengzhou University, Zhengzhou, China

* zdong39@zzu.edu.cn

## Abstract

Multivariate statistical techniques, including cluster analysis (CA), discriminant analysis (DA), principal component analysis (PCA) and factor analysis (FA), were used to evaluate temporal and spatial variations in and to interpret large and complex water quality datasets collected from the Shuangji River Basin. The datasets, which contained 19 parameters, were generated during the 2 year (2018–2020) monitoring programme at 14 different sites (3192 observations) along the river. Hierarchical CA was used to divide the twelve months into three periods and the fourteen sampling sites into three groups. Discriminant analysis identified four parameters (CODMn, Cu, As, Se) loading more than 68% correct assignations in temporal analysis, while seven parameters (COD, TP, CODMn, F, LAS, Cu and Cd) to load 93% correct assignations in spatial analysis. The FA/PCA identified six factors that were responsible for explaining the data structure of 68% of the total variance of the dataset, allowing grouping of selected parameters based on common characteristics and assessing the incidence of overall change in each group. This study proposes the necessity and practicality of multivariate statistical techniques for evaluating and interpreting large and complex data sets, with a view to obtaining better information about water quality and the design of monitoring networks to effectively manage water resources.

## Introduction

Water is the material basis for the existence of earth creatures, and water resources are the primary condition for maintaining the sustainable development of the earth's ecological environment [1]. With the increasing consumption of water resources, the contradiction between the supply and demand of water resources has intensified, which puts forward greater requirements for the utilization and protection of surface water resources [2].

The surface water quality of a region depends to a large extent on environmental factors (temperature changes, precipitation and soil erosion) and human input (discharge of municipal and industrial wastewater and over-exploitation of water resources) [3]. Among them, the discharge of

**Funding:** This study was supported by Major Science and Technology Program for Water Pollution Control and Treatment of China (No. 2017ZX07602-001-002).

**Competing interests:** We declare that we have no financial and personal relationships with other people or organizations that can inappropriately influence our work, there is no professional or other personal interest of any nature or kind in any product, service and/or company that could be construed as influencing the position presented in, or the review of, the manuscript entitled.

urban sewage and industrial wastewater is a continuous source of pollution, so effective control of sewage discharge is of great significance to the improvement of water quality [4, 5].

Surface water runoff is a seasonal phenomenon that is mainly affected by the climate of the basin [6]. In addition, seasonal changes in precipitation, surface runoff, interflow, groundwater flow, and pumping in and pumping out have a strong influence on the river flow and the subsequent pollutant concentration in the river [7]. Therefore, correct identification of potential sources of surface water quality pollution is the basis and prerequisite for water quality management.

Shuangji River is a polluted river. Its main source of water comes from urban sewage treatment plants and paper-making sewage treatment plants. It not only plays an important role in assimilating or removing urban and industrial wastewater and farmland runoff, but is also the main inland water resources used for household, industrial, and irrigation purposes [8], Therefore, it is necessary to prevent and control river pollution and have reliable water quality information for effective management. Given the spatial and temporal changes in river water chemistry, regular monitoring programmes are needed to reliably estimate water quality [9]. This leads large and complex data matrices composed of a large number of physical and chemical parameters, which are often difficult to interpret, making it challenging to draw meaningful conclusions [10].

Multivariate statistical analysis is a branch developed from classical statistics and is a comprehensive analysis method [11, 12]. It can analyze the statistical laws of multiple objects and multiple indicators when they are related to each other, including cluster analysis (CA) [13], discriminant analysis (DA) [14], principal component analysis (PCA) [15] and factor analysis (FA) [16]. Multivariate statistical analysis is a suitable tool for multi-component chemical and physical measurements for meaningful data reduction and interpretation [17]. It is a valuable tool for identifying factors and sources that may affect water systems and cause changes in water quality [18].

In this article, we took the Shuangji River as the research object for the first time, set up 14 main detection points along the river and detected and analyzed 19 physical and chemical parameters in water samples. The detection time lasted for 2 years. Different multivariate statistical techniques were used to analyse the obtained datasets, to analyse the similarity or dissimilarity between monitoring periods or monitoring points, to identify the water quality variables that cause the spatial and temporal changes of river water quality, and to determine the impact of water sources (natural and anthropogenic factors).

## Materials and methods

### Study area

The Shuangji River (N-34°22′-34°30′, E-113°13′-113°37′), a tributary of the Huai River, originates from the eastern side of Wuzhiling in northwestern Xinmi County and flows through 57 administrative villages in 8 towns in Xinmi County, covering 57 kilometres and controlling the Xinmi River Basin, which has an area of 868 km$^2$ (Fig 1). The Weishui River (T1), Zhaoyangshui River (T2), Liquan River (T3), Yang River (T4), Ze River (T5), and Wu River (T6) are the main branches of the Shuangji River (Fig 1). The three sides of Xinmi County are located in the eastern part of Henan Province, which is surrounded by mountains on three sides. The terrain is high in the west and low in the east. It is a closed watershed with no external water supply. The average annual rainfall is 660 mm. The Shuangji River is basically free of external runoff recharge, and the main body of the river is affected by domestic sewage and industrial wastewater (Shuangji River is an open river and can be used without any permit. Anyone can study and use the Shuangji River. Therefore, the study of the Shuangji River in this article does not require a permit).

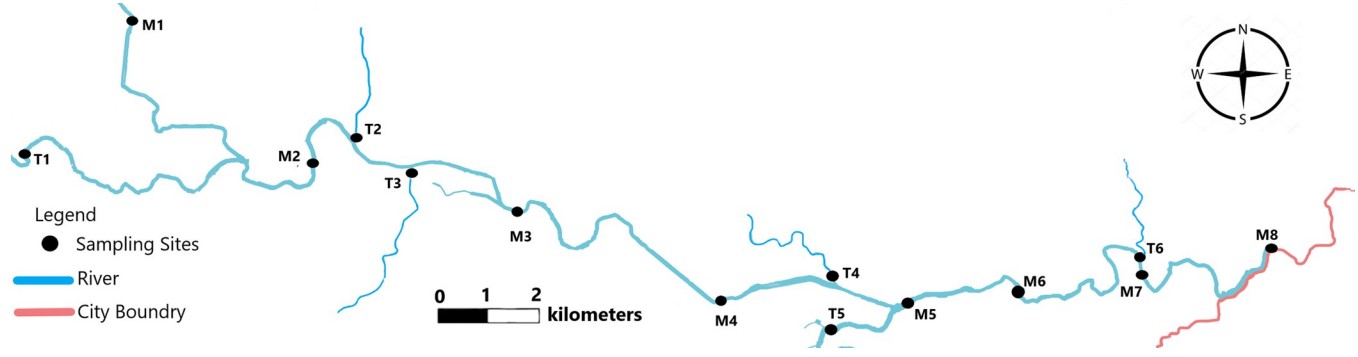

**Fig 1. Map of study area and water quality monitoring sites.**

### Sampling collection and pretreatment

The design of the sampling network covers the identification of a wide range of key locations, including tributaries and water inputs that have a greater impact on the river [19], and reasonably represent the water quality of the river system (Fig 1). Sites T1, T2, T3, T4, T5 and T6 are the main tributaries entering the Shuangji River. Sites M1, M2, M3, M4, M5, M6, M7 and M8 are the key points on the main bank of the Shuangji River, where site M8 is under municipal control and provides the exit water of Xinmi County. Sites M1-M3 and T1-T3 are closer to the urban area, and the main source of pollution is urban sewage entering the river. Sites M4, M5, T4 and T5 are located in the industrial zone, and the paper industry in their vicinity is relatively developed; as a result, the pollution is mainly due to paper industry wastewater entering the river. Sites M6-M8 are downstream of the river and have no inflow of external water.

The dataset included 19 water quality parameters that were monitored by sampling at 14 monitoring points for 2 years (2018–2020). The factors monitored in this study included pH, dissolved oxygen (DO), chemical oxygen demand (COD), ammonia-nitrogen ($NH_3$-N), total phosphorus (TP), chemical oxygen demand (CODMn), fluoride (F), petroleum hydrocarbons (oil), linear alkylbenzene sulfonates (LAS), copper (Cu), zinc (Zn), cadmium (Cd), arsenic (As), mercury (Hg), hexavalent chromium ($Cr^{6+}$), total cyanides (CN), volatile phenols (VP), sulphide (S) and selenium (Se). When collecting water quality samples, a standard open barrel sampler (1.5 litre capacity) was used to collect water samples. This sampler can collect water samples from different depths of water to ensure the representativeness of the data. Before collecting the water sample, the 2 litre polyethylene plastic bottle was washed with metal-free soap, rinsed several times with distilled water, soaked in 10% nitric acid for 24 hours, and finally rinsed with ultrapure water for sampling. All water samples collected were first stored in an insulated cooler and placed in a refrigerator at 4˚C and sent to the laboratory for analysis on the day the water samples were collected.

### Analytical procedure

The water quality parameters, analytical units and analytical methods are summarized in Table 1. The pH value and DO value of each water sample were determined on site using digital pH values (JY-PH6.0) and DO measuring instruments (YT-RJY). Water samples of approximately 1000 mL were taken at each sampling point in the field and filtered through a polycarbonate filter (0.45 μm pore size). The pretreatment of the sample was divided into two parts. One part of the sample was used for physical and chemical parameters and anion analysis and was directly tested, while the other part was first treated with 2 mL of concentrated $HNO_3$

Table 1. Water quality parameters associated with their acronyms, units and analytical methods used.

| S.N. | Variables | Acronyms | Units | Analytical methods |
|------|-----------|----------|-------|--------------------|
| 1 | pH | pH | pH unit | pH meter |
| 2 | Dissolved oxygen | DO | mg/L | Prob method |
| 3 | Chemical oxygen demand | COD | mg/L | Dichromate method |
| 4 | Ammonical nitrogen | $NH_3$-N | mg/L | Spectrophotometric |
| 5 | Total phosphorus | TP | mg/L | Ammonium molybdate Spectrophotometry |
| 6 | Chemical oxygen demand(Mn) | CODMn | mg/L | Permanganate index method |
| 7 | Fluoride | F | mg/L | Ion selective electrode |
| 8 | Petroleum oil | Oil | mg/L | Infrared spectrophotometry |
| 9 | Linear alkylbenzene sulfonates | LAS | mg/L | Methylene blue Spectrophotometry |
| 10 | Copper | Cu | mg/L | FAAS |
| 11 | Zinc | Zn | mg/L | FAAS |
| 12 | Cadmium | Cd | mg/L | ETAAS |
| 13 | Arsenic | As | mg/L | HGAAS |
| 14 | Mercury | Hg | mg/L | CVAAS |
| 15 | Hexavalent chromium | $Cr^{6+}$ | mg/L | ETAAS |
| 16 | Total cyanide | CN | mg/L | Pyridine barbituric acid Spectrophotometry |
| 17 | Volatile phenol | VP | mg/L | Spectrophotometric Determination with 4-Amino-Antipyrin |
| 18 | Sulfide | S | mg/L | Methylene blue Spectrophotometry |
| 19 | Selenium | Se | mg/L | HGAAS |

before being subjected to metal analysis. All samples were analysed within 48 hours. The COD was measured by the dichromate reflux method (DH310C1COD) [20], and $NH_3$-N was measured with Nessler's reagent (NH3N-1040) [21]. The TP was measured by ammonium molybdate spectrophotometry (HM-812) [22], and the CODMn was measured by the permanganate index method (Thermo Scientific 3131) [23]. Fluoride (F) was measured using an ion-selective electrode (BHF5300) [24], and the total cyanides were analysed using pyridine barbituric acid spectrophotometry (TCN-508) [25], while sulphide (S) was determined using methylene blue spectrophotometry (ST201A) [26]. Petroleum hydrocarbons (Oil) were measured using infrared spectrophotometry (GC1290) [27]. The linear alkylbenzene sulfonates (LAS) were measured using methylene blue spectrophotometry (UltiMate3000) [28], and volatile phenols (VP) were measured using spectrophotometric determination with 4-amino-antipyrin (BELL) [29].

The main cation was determined by subjecting the acid-treated water samples to a 20-fold dilution with ultrapure water. For the trace elements and toxic elements, the volume of the water samples was reduced by a factor of four at 60°C on an electric hot plate. Cu and Zn were determined by a flame atomic absorption spectrometer (FAAS) using an ethane-air flame (CAAM-2001N) [30], while Hg was determined by cold-vapor atomic absorption spectrometry (CVAAS) (Ultima Expert) [31]. Cd and Cr were measured using an electrothermal atomic absorption spectrometer (ETAAS) (Avio 200) [32], while As and Se were analysed using the hydride generation method (HGAAS) (AA-6033C) [33]. The accuracy of the analytical data were ensured by triplicate samples, blank test controls and careful standardization. The ion balance of each sample was within ±5%.

## Data treatment and multivariate statistical methods

Although water sampling was conducted every month at all sites, due to the impact of the COVID-19 pandemic and bad weather, some points could not be sampled, and the missing data were replaced by the average value. The basic statistics of the two-year water quality

dataset (3192 observations) are shown in Table 2. The data for multivariate statistical analysis usually conform to a normal distribution; therefore, before conducting the multivariate statistical analysis, each variable was tested for conformity to the normal distribution by analysing the skewness and kurtosis statistics. The test results showed that all factors were in line with or close to the normal distribution. The ranges of skewness and kurtosis were—0.45 to 0.91 and —0.97 to 0.53, respectively. For CA and PCA, taking into account the differences in the magnitude and measurement units of different water quality indicators, all selected parameters were also z-scale normalized with mean = 1 and variance = 0.

In this study, all data were analysed through a variety of multivariate statistical analysis techniques to explore the parameters that caused changes in water quality at different temporal and spatial scales [34]. For effective pollution control and water management, a large amount of water quality data needs to be explained. Controlling river pollution and mastering reliable water quality information are necessary for effective management. Multivariate analysis of river water quality datasets by CA, DA, PCA and FA, CA, PCA, and FA were applied to experimental data and normalized by z-scale conversion to avoid misclassification due to large differences in data dimensions, and DA was applied to the original data. All mathematical and statistical calculations were performed using Excel 2010, IBM SPSS Statistics 26.0 and Statistica 12 [35–37].

## Cluster analysis

Cluster analysis (CA) is a multivariate statistical method for classifying objects according to their distance or proximity [38]. The system objects can be classified into categories or clusters based on the similarity or difference of their objects [39]. The hierarchical CA method adopted in this paper is the most widely used clustering method. This method clusters the closest or most similar objects into clusters through successive aggregation and finally groups these clusters into larger clusters. The Euclidean distance usually indicates whether two samples are similar, and the "distance" can be expressed by the "difference" between the analysis values of the two samples [40]. In this study, the Ward method was used with the squared Euclidean distance as a measure of similarity, and a hierarchical aggregate CA was performed on a normalized dataset. The distance between clusters was determined using analysis of variance, and the sum of squares of the two clusters generated in each step was minimized. CA analyses river water quality datasets to group spatial and temporal variability by similarities, thereby creating a spatiotemporal tree among samples. The dendrogram provides a visual summary of the clustering process, showing an image of each group and those in its vicinity, while the dimensions of the original data are greatly reduced. The link distance is reported as Dlink/Dmax, which represents the quotient of the link distance divided by the maximum distance and multiplied by 100 in a specific case to standardize the link distance on the y-axis. The standardized data were clustered by the Ward method and square Euclidean distance [41].

## Discriminant analysis

Discriminant analysis (DA) is used to analyse the difference between two or more naturally occurring groups [42]. It can establish a discriminant function when the previous class is known and assign observations to known groups. If the DA is valid for a set of data, a correctly and incorrectly estimated classification table will produce a high correct percentage. DA distinguishes between two or more naturally occurring groups by quantitative attributes and aims to provide a statistical classification of samples, which can be performed by CA. The DA technique establishes a discriminant function for each group, which operates based on the

**Table 2. Statistical description (max, min, mean and SD) of water quality parameters.**

| Parameters | | M1 | T1 | M2 | T2 | T3 | M3 | M4 | T4 | T5 | M5 | M6 | T6 | M7 | M8 |
|---|---|---|---|---|---|---|---|---|---|---|---|---|---|---|---|
| pH | Max. | 7.81 | 8.31 | 7.99 | 7.84 | 8.05 | 7.98 | 7.98 | 7.98 | 7.93 | 7.96 | 7.82 | 7.81 | 7.89 | 7.97 |
| | Min. | 7.37 | 7.31 | 7.40 | 7.27 | 7.28 | 7.32 | 7.31 | 7.32 | 7.29 | 7.28 | 7.32 | 7.32 | 7.38 | 7.41 |
| | Mean | 7.56 | 7.65 | 7.63 | 7.60 | 7.63 | 7.79 | 7.70 | 7.64 | 7.56 | 7.79 | 7.61 | 7.62 | 7.62 | 7.70 |
| | SD | 0.17 | 0.25 | 0.22 | 0.19 | 0.20 | 0.19 | 0.23 | 0.21 | 0.27 | 0.23 | 0.18 | 0.17 | 0.20 | 0.23 |
| DO (mg/L) | Max. | 8.50 | 9.10 | 9.00 | 8.90 | 9.20 | 9.60 | 9.40 | 8.90 | 8.80 | 9.70 | 9.10 | 8.80 | 9.10 | 9.40 |
| | Min. | 6.80 | 8.40 | 7.90 | 7.70 | 8.20 | 7.80 | 8.10 | 7.60 | 7.60 | 7.10 | 7.60 | 8.20 | 7.60 | 7.80 |
| | Mean | 7.57 | 8.71 | 8.56 | 8.23 | 8.63 | 8.53 | 8.58 | 8.41 | 8.36 | 8.09 | 8.18 | 8.52 | 8.38 | 8.85 |
| | SD | 0.63 | 0.18 | 0.37 | 0.33 | 0.34 | 0.55 | 0.47 | 0.49 | 0.40 | 0.89 | 0.51 | 0.19 | 0.48 | 0.57 |
| COD (mg/L) | Max. | 36 | 34 | 42 | 16 | 15 | 24 | 32 | 19 | 24 | 44 | 42 | 36 | 30 | 29 |
| | Min. | 23 | 6 | 6 | 4 | 5 | 6 | 15 | 5 | 6 | 14 | 13 | 9 | 16 | 15 |
| | Mean | 30.08 | 15.67 | 26.92 | 9.17 | 8.83 | 17.58 | 26.58 | 11.17 | 12.50 | 32.08 | 26.33 | 22.17 | 25.42 | 23.58 |
| | SD | 4.12 | 9.12 | 9.90 | 3.86 | 3.41 | 5.95 | 5.09 | 5.59 | 5.63 | 9.56 | 8.77 | 9.16 | 4.56 | 4.29 |
| NH$_3$-N(mg/L) | Max. | 2.22 | 1.68 | 1.63 | 1.82 | 1.89 | 1.90 | 1.78 | 0.97 | 0.87 | 1.22 | 1.64 | 1.42 | 1.24 | 0.98 |
| | Min. | 1.02 | 0.42 | 1.23 | 0.86 | 0.82 | 0.79 | 0.62 | 0.33 | 0.39 | 0.56 | 0.52 | 0.46 | 0.44 | 0.45 |
| | Mean | 1.64 | 0.89 | 1.39 | 1.34 | 1.46 | 1.27 | 1.20 | 0.62 | 0.66 | 0.93 | 0.90 | 0.91 | 0.86 | 0.74 |
| | SD | 0.32 | 0.38 | 0.13 | 0.29 | 0.35 | 0.33 | 0.36 | 0.22 | 0.15 | 0.23 | 0.34 | 0.28 | 0.28 | 0.15 |
| TP (mg/L) | Max. | 0.44 | 0.32 | 0.38 | 0.30 | 0.20 | 0.36 | 0.32 | 0.23 | 0.15 | 0.33 | 0.19 | 0.32 | 0.28 | 0.26 |
| | Min. | 0.31 | 0.13 | 0.20 | 0.08 | 0.05 | 0.12 | 0.12 | 0.14 | 0.03 | 0.11 | 0.10 | 0.17 | 0.07 | 0.08 |
| | Mean | 0.38 | 0.20 | 0.31 | 0.16 | 0.10 | 0.22 | 0.23 | 0.19 | 0.10 | 0.19 | 0.14 | 0.22 | 0.16 | 0.15 |
| | SD | 0.05 | 0.05 | 0.06 | 0.07 | 0.04 | 0.07 | 0.06 | 0.03 | 0.04 | 0.08 | 0.03 | 0.05 | 0.07 | 0.05 |
| CODMn (mg/L) | Max. | 9.90 | 4.80 | 8.20 | 5.90 | 6.10 | 6.80 | 7.50 | 4.90 | 3.90 | 8.20 | 8.00 | 8.50 | 7.80 | 8.30 |
| | Min. | 7.40 | 2.60 | 6.20 | 3.40 | 5.10 | 4.60 | 4.80 | 2.50 | 2.40 | 3.90 | 3.90 | 5.10 | 4.60 | 4.20 |
| | Mean | 8.28 | 3.73 | 7.09 | 4.69 | 5.59 | 6.07 | 6.16 | 3.59 | 3.21 | 6.53 | 5.84 | 6.65 | 6.38 | 6.34 |
| | SD | 0.83 | 0.68 | 0.70 | 0.84 | 0.38 | 0.70 | 1.03 | 0.99 | 0.52 | 1.39 | 1.61 | 1.34 | 1.25 | 1.38 |
| F (mg/L) | Max. | 0.75 | 0.94 | 1.10 | 0.48 | 0.52 | 0.64 | 0.77 | 0.71 | 0.63 | 0.76 | 0.90 | 0.98 | 0.86 | 0.85 |
| | Min. | 0.44 | 0.72 | 0.75 | 0.38 | 0.38 | 0.52 | 0.58 | 0.51 | 0.53 | 0.62 | 0.73 | 0.72 | 0.71 | 0.66 |
| | Mean | 0.55 | 0.85 | 0.84 | 0.44 | 0.45 | 0.61 | 0.64 | 0.61 | 0.59 | 0.67 | 0.81 | 0.87 | 0.81 | 0.78 |
| | SD | 0.12 | 0.07 | 0.13 | 0.03 | 0.07 | 0.03 | 0.06 | 0.08 | 0.03 | 0.05 | 0.06 | 0.08 | 0.05 | 0.08 |
| Oil (mg/L) | Max. | 0.24 | 0.18 | 0.20 | 0.09 | 0.09 | 0.07 | 0.17 | 0.11 | 0.06 | 0.12 | 0.21 | 0.18 | 0.19 | 0.18 |
| | Min. | 0.13 | 0.06 | 0.05 | 0.05 | 0.05 | 0.05 | 0.06 | 0.06 | 0.02 | 0.06 | 0.06 | 0.06 | 0.06 | 0.06 |
| | Mean | 0.17 | 0.10 | 0.11 | 0.08 | 0.07 | 0.06 | 0.09 | 0.08 | 0.05 | 0.08 | 0.15 | 0.15 | 0.14 | 0.14 |
| | SD | 0.03 | 0.04 | 0.06 | 0.01 | 0.01 | 0.01 | 0.03 | 0.02 | 0.02 | 0.02 | 0.05 | 0.04 | 0.04 | 0.04 |
| Cr$^{6+}$ (mg/L) | Max. | 0.023 | 0.013 | 0.024 | 0.023 | 0.019 | 0.021 | 0.022 | 0.018 | 0.015 | 0.026 | 0.023 | 0.024 | 0.023 | 0.022 |
| | Min. | 0.013 | 0.008 | 0.015 | 0.007 | 0.008 | 0.012 | 0.015 | 0.010 | 0.006 | 0.012 | 0.016 | 0.017 | 0.015 | 0.015 |
| | Mean | 0.018 | 0.010 | 0.019 | 0.015 | 0.015 | 0.015 | 0.019 | 0.014 | 0.009 | 0.018 | 0.021 | 0.020 | 0.019 | 0.017 |
| | SD | 0.003 | 0.002 | 0.002 | 0.005 | 0.004 | 0.003 | 0.002 | 0.002 | 0.003 | 0.005 | 0.002 | 0.002 | 0.002 | 0.002 |
| LAS (mg/L) | Max. | 0.30 | 0.29 | 0.32 | 0.23 | 0.23 | 0.23 | 0.28 | 0.24 | 0.21 | 0.26 | 0.29 | 0.27 | 0.28 | 0.26 |
| | Min. | 0.16 | 0.07 | 0.11 | 0.09 | 0.08 | 0.07 | 0.09 | 0.05 | 0.04 | 0.08 | 0.07 | 0.08 | 0.07 | 0.06 |
| | Mean | 0.27 | 0.24 | 0.24 | 0.17 | 0.16 | 0.18 | 0.22 | 0.17 | 0.15 | 0.19 | 0.21 | 0.22 | 0.21 | 0.20 |
| | SD | 0.04 | 0.07 | 0.08 | 0.05 | 0.04 | 0.05 | 0.06 | 0.07 | 0.05 | 0.07 | 0.08 | 0.05 | 0.06 | 0.06 |
| Cu (mg/L) | Max. | 0.07 | 0.08 | 0.08 | 0.06 | 0.05 | 0.06 | 0.06 | 0.04 | 0.05 | 0.06 | 0.06 | 0.08 | 0.06 | 0.05 |
| | Min. | 0.04 | 0.03 | 0.06 | 0.03 | 0.02 | 0.03 | 0.03 | 0.02 | 0.02 | 0.03 | 0.02 | 0.03 | 0.02 | 0.02 |
| | Mean | 0.06 | 0.05 | 0.07 | 0.04 | 0.03 | 0.04 | 0.04 | 0.03 | 0.03 | 0.04 | 0.04 | 0.05 | 0.04 | 0.03 |
| | SD | 0.01 | 0.01 | 0.01 | 0.01 | 0.01 | 0.01 | 0.01 | 0.01 | 0.01 | 0.01 | 0.01 | 0.02 | 0.01 | 0.01 |
| Zn (mg/L) | Max. | 0.06 | 0.05 | 0.06 | 0.06 | 0.06 | 0.07 | 0.06 | 0.06 | 0.07 | 0.06 | 0.06 | 0.06 | 0.05 | 0.05 |
| | Min. | 0.04 | 0.03 | 0.03 | 0.03 | 0.02 | 0.03 | 0.03 | 0.03 | 0.03 | 0.03 | 0.03 | 0.03 | 0.03 | 0.03 |
| | Mean | 0.05 | 0.04 | 0.04 | 0.04 | 0.04 | 0.05 | 0.04 | 0.04 | 0.04 | 0.04 | 0.04 | 0.05 | 0.04 | 0.04 |
| | SD | 0.01 | 0.01 | 0.01 | 0.01 | 0.01 | 0.01 | 0.01 | 0.01 | 0.01 | 0.01 | 0.01 | 0.01 | 0.01 | 0.01 |

(*Continued*)

**Table 2.** (Continued)

| Parameters | | M1 | T1 | M2 | T2 | T3 | M3 | M4 | T4 | T5 | M5 | M6 | T6 | M7 | M8 |
|---|---|---|---|---|---|---|---|---|---|---|---|---|---|---|---|
| Cd (mg/L) | Max. | 0.00033 | 0.00029 | 0.00033 | 0.00028 | 0.00026 | 0.00036 | 0.00039 | 0.00036 | 0.00028 | 0.00048 | 0.00048 | 0.00042 | 0.00038 | 0.00033 |
| | Min. | 0.00018 | 0.00012 | 0.00019 | 0.00022 | 0.00016 | 0.00025 | 0.00021 | 0.00012 | 0.00012 | 0.00022 | 0.00024 | 0.00026 | 0.00020 | 0.00016 |
| | Mean | 0.00024 | 0.00021 | 0.00027 | 0.00025 | 0.00021 | 0.00030 | 0.00030 | 0.00029 | 0.00021 | 0.00034 | 0.00035 | 0.00034 | 0.00029 | 0.00026 |
| | SD | 0.00004 | 0.00006 | 0.00005 | 0.00002 | 0.00003 | 0.00003 | 0.00007 | 0.00007 | 0.00004 | 0.00008 | 0.00009 | 0.00006 | 0.00006 | 0.00005 |
| As (mg/L) | Max. | 0.0036 | 0.0034 | 0.0036 | 0.0028 | 0.0022 | 0.0038 | 0.0036 | 0.0032 | 0.0026 | 0.0036 | 0.0035 | 0.0037 | 0.0032 | 0.0030 |
| | Min. | 0.0016 | 0.0008 | 0.0015 | 0.0008 | 0.0008 | 0.0011 | 0.0015 | 0.0008 | 0.0008 | 0.0012 | 0.0009 | 0.0012 | 0.0008 | 0.0007 |
| | Mean | 0.0029 | 0.0022 | 0.0025 | 0.0017 | 0.0016 | 0.0020 | 0.0027 | 0.0020 | 0.0018 | 0.0029 | 0.0027 | 0.0030 | 0.0026 | 0.0024 |
| | SD | 0.0006 | 0.0009 | 0.0008 | 0.0006 | 0.0004 | 0.0009 | 0.0006 | 0.0006 | 0.0005 | 0.0007 | 0.0007 | 0.0007 | 0.0007 | 0.0007 |
| Hg (mg/L) | Max. | 0.00026 | 0.00025 | 0.00026 | 0.00012 | 0.00030 | 0.00026 | 0.00023 | 0.00016 | 0.00016 | 0.00019 | 0.00016 | 0.00018 | 0.00016 | 0.00021 |
| | Min. | 0.00010 | 0.00007 | 0.00006 | 0.00008 | 0.00012 | 0.00009 | 0.00012 | 0.00006 | 0.00005 | 0.00012 | 0.00007 | 0.00008 | 0.00007 | 0.00005 |
| | Mean | 0.00017 | 0.00015 | 0.00015 | 0.00010 | 0.00018 | 0.00016 | 0.00016 | 0.00011 | 0.00010 | 0.00015 | 0.00013 | 0.00015 | 0.00012 | 0.00011 |
| | SD | 0.00006 | 0.00008 | 0.00008 | 0.00001 | 0.00006 | 0.00005 | 0.00004 | 0.00003 | 0.00004 | 0.00002 | 0.00003 | 0.00003 | 0.00003 | 0.00005 |
| CN (mg/L) | Max. | 0.019 | 0.018 | 0.016 | 0.009 | 0.009 | 0.014 | 0.013 | 0.013 | 0.013 | 0.019 | 0.018 | 0.009 | 0.017 | 0.013 |
| | Min. | 0.012 | 0.005 | 0.011 | 0.006 | 0.005 | 0.006 | 0.006 | 0.005 | 0.004 | 0.007 | 0.005 | 0.006 | 0.005 | 0.004 |
| | Mean | 0.015 | 0.012 | 0.013 | 0.008 | 0.007 | 0.008 | 0.010 | 0.009 | 0.008 | 0.014 | 0.012 | 0.008 | 0.011 | 0.009 |
| | SD | 0.003 | 0.005 | 0.002 | 0.001 | 0.001 | 0.002 | 0.002 | 0.003 | 0.003 | 0.004 | 0.005 | 0.001 | 0.004 | 0.003 |
| VP (mg/L) | Max. | 0.0015 | 0.0009 | 0.0014 | 0.0009 | 0.0007 | 0.0012 | 0.0016 | 0.0009 | 0.0008 | 0.0013 | 0.0013 | 0.0009 | 0.0013 | 0.0010 |
| | Min. | 0.0008 | 0.0003 | 0.0006 | 0.0003 | 0.0003 | 0.0004 | 0.0008 | 0.0004 | 0.0003 | 0.0004 | 0.0003 | 0.0003 | 0.0004 | 0.0003 |
| | Mean | 0.0011 | 0.0006 | 0.0010 | 0.0006 | 0.0005 | 0.0007 | 0.0011 | 0.0006 | 0.0005 | 0.0006 | 0.0007 | 0.0007 | 0.0008 | 0.0006 |
| | SD | 0.0002 | 0.0002 | 0.0002 | 0.0002 | 0.0001 | 0.0002 | 0.0003 | 0.0002 | 0.0002 | 0.0003 | 0.0004 | 0.0002 | 0.0003 | 0.0003 |
| S (mg/L) | Max. | 0.014 | 0.014 | 0.015 | 0.008 | 0.010 | 0.009 | 0.013 | 0.009 | 0.009 | 0.009 | 0.007 | 0.010 | 0.009 | 0.007 |
| | Min. | 0.010 | 0.006 | 0.009 | 0.006 | 0.005 | 0.005 | 0.009 | 0.005 | 0.005 | 0.006 | 0.005 | 0.006 | 0.006 | 0.005 |
| | Mean | 0.012 | 0.010 | 0.012 | 0.007 | 0.008 | 0.007 | 0.011 | 0.007 | 0.006 | 0.007 | 0.006 | 0.008 | 0.007 | 0.006 |
| | SD | 0.001 | 0.003 | 0.003 | 0.001 | 0.001 | 0.002 | 0.001 | 0.001 | 0.001 | 0.001 | 0.001 | 0.001 | 0.001 | 0.001 |
| Se (mg/L) | Max. | 0.0013 | 0.0009 | 0.0012 | 0.0009 | 0.0007 | 0.0013 | 0.0016 | 0.0009 | 0.0008 | 0.0014 | 0.0009 | 0.0013 | 0.0010 | 0.0009 |
| | Min. | 0.0006 | 0.0004 | 0.0006 | 0.0004 | 0.0003 | 0.0004 | 0.0004 | 0.0005 | 0.0004 | 0.0004 | 0.0004 | 0.0005 | 0.0004 | 0.0004 |
| | Mean | 0.0009 | 0.0007 | 0.0009 | 0.0006 | 0.0005 | 0.0007 | 0.0010 | 0.0006 | 0.0006 | 0.0009 | 0.0007 | 0.0009 | 0.0007 | 0.0007 |
| | SD | 0.0003 | 0.0001 | 0.0002 | 0.0002 | 0.0002 | 0.0003 | 0.0005 | 0.0002 | 0.0002 | 0.0003 | 0.0002 | 0.0003 | 0.0002 | 0.0002 |

original data [43], as shown below:

$$f(\mathrm{G_i}) = k_i + \sum_{j=1}^{n} w_{ij} p i_j \tag{1}$$

Where $i$ is the number of groups (G), $k_i$ is the constant inherent to each group, $n$ is the number of parameters used to classify a set of data into a given group, and $w_j$ is the weight coefficient that is assigned by DA to a given selected parameter ($p_j$).

In this study, three groups of temporal (three seasons) and spatial (three sampling areas) evaluations were selected, and the analysis parameters used to assign the measurement of one monitoring point to one group (season or monitoring area) were taken as n. The discriminant analysis of the original data was run in standard mode, forward stepwise mode and backward stepwise mode to construct discriminant functions to evaluate the temporal and spatial changes in river water quality. The site (spatial) and season (temporal) are group-dependent variables, and all measurement parameters are independent variables.

### Principal component analysis (PCA)/factor analysis (FA)

Principal component analysis (PCA) is actually a dimensionality reduction method. Its main purpose is to use fewer variables to explain most of the variation in the original data and to convert many highly correlated variables into independent or unrelated variables [44]. Usually, new variables that are fewer in number than the original variables and that can explain the variation in most of the data, the so-called principal components, are used to explain the comprehensive index of the data. The basic idea of principal component analysis is to first draw a "best" fitting line for n points so that the sum of squares of the vertical distance of these n points to the line is the smallest and is called the first principal component of this line [45]. Then, the second principal component that is independent of the first principal component and has the smallest square sum of vertical distances from n points is found. Analogously, until m principal components are obtained, the value of m is usually such that the variance of the first few principal components accounts for more than 85% of the total variance [46].

Factor analysis (factor analysis) is a multivariate statistical method that uses a few potential random variables—factors—to describe the covariance relationship among many variables [47]. In this paper, the factor obtained by the rotation of the maximum variance criterion is a linear combination of the original variables [48]. Under the premise of ensuring the least information loss, the original data are described as accurately as possible to achieve the dimensionality reduction of multivariate data. In general, the analysis results only select factors with eigenvalues greater than 1.

## Results and discussion

### Temporal/spatial similarities and grouping

The dendrogram generated by the time cluster analysis divided 12 months into three clusters at (Dlink/Dmax) *100 <70, and there were significant differences between the clusters (Fig 2). The first cluster (first period) included June and July, corresponding to the high water flow period; the second cluster (2nd period) included August, September, October and November,

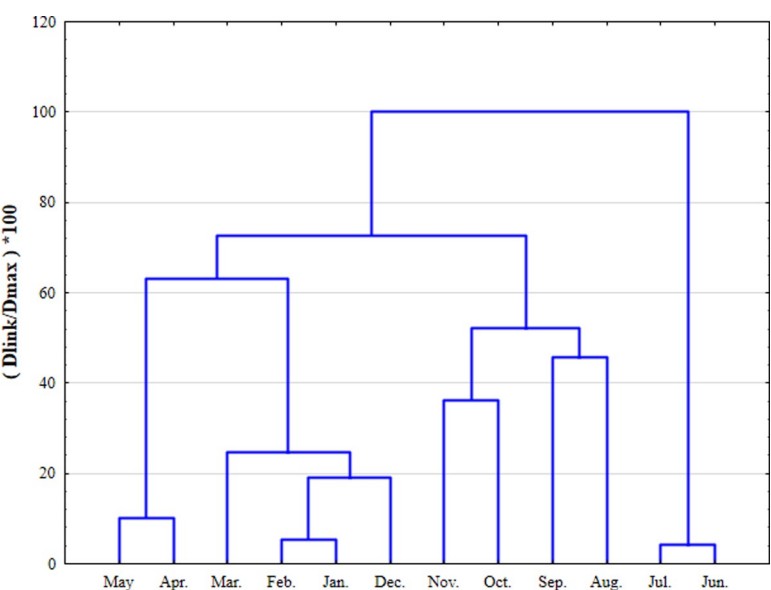

**Fig 2. Dendrogram showing temporal clustering of monitoring periods.**

corresponding to the flat water flow period; the third cluster (3rd period) contained all the remaining months (December, January to May), corresponding to the low water flow period. Therefore, the temporal change in river water quality depends largely on local climatic conditions (spring, summer, autumn, winter) and hydrological conditions (low flow, average flow, and high flow periods). Obviously, the Shuangji River Basin is a typical seasonal river in North China. Since the Shuangji River is mainly a polluted river, the main body of the river comes from the sewage treatment plant along the bank, and the change in water quality reflects the change in the treatment effect of the sewage treatment plant. In summer, the sewage treatment plant has a better treatment effect, and the summer rainfall is large, and the river flow is large, so the river water quality in summer is better and divided into one category. In winter, the sewage treatment plant has poor water quality due to temperature and operation, and the rainfall is small, and the river flow is small. Therefore, the river water quality in winter is poor and divided into one category.

The spatial CA also generated a dendrogram with three clusters at (Dlink/Dmax)*100<50 (Fig 3). Group A comprised M1 and M2; group B comprised T6 and M3 to M8; group C comprised T1 to T5. It can be clearly seen from the Fig 3 that one group was the main branch of river (M1 to M8), while the other type was the tributaries of the river (T1 to T6). The tributary water sources of the Shuangji River mainly come from the drainage of upstream coal mines and reservoirs. Compared with the main river, the tributary water sources are very clean and were therefore classified as a cluster. The main river category was divided into three categories at (Dlink/Dmax)*100<30. The first category included M1 and M2, which were highly polluted areas. The second category included M3, M4 and M5, which were in moderately polluted areas. Among the pollution sources, the main source of pollution in the high-pollution areas was that the surrounding rural domestic sewage was directly discharged into rivers and urban sewage after treatment by sewage treatment plants. Due to the improvement in urban living standards, urban domestic water consumption has exceeded the carrying capacity of sewage treatment plants, and a new sewage treatment plant is currently under construction, resulting in poor water quality in the upper Shuangji River. The medium-pollution areas were mainly polluted by the industrial wastewater discharged into the river (It can also be known from the

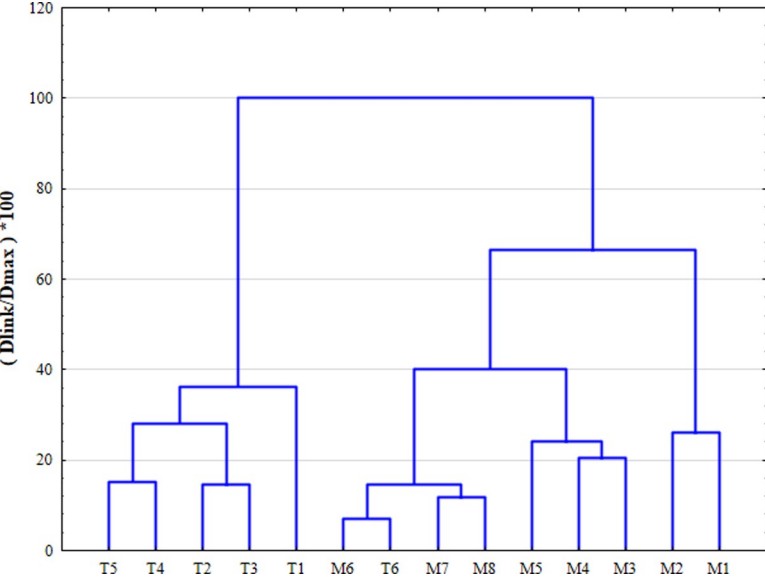

**Fig 3. Dendrogram showing spatial clustering of monitoring sites.**

data in Table 2). The main discharge enterprise was the paper factory, and the main pollution factor was COD. However, due to the recent strict national requirements for wastewater discharge, the wastewater of the paper mill has been discharged through the factory to meet the standard discharge, which has not caused much river pollution, resulting in the areas being only moderately polluted. The low-pollution area was located in the lower reaches of the Shuangji River in Xinmi County. There was no external water pollution. The river has passed the purification of constructed wetlands and its own self-purification ability to achieve better water quality, so it belongs to the low-pollution area.

## Temporal/spatial variations in river water quality

The temporal variation was evaluated using DA, with the clusters based on CA. DA aims to test the importance of discriminant functions and to determine the most important variables related to the differences between clusters. As shown in Table 3, the Wilks' lambda and chi-square values of each discriminant function ranged from 0.273 to 0.808 and from 34.834 to 202.505. The p-level value was lower than 0.01, indicating that the time DA was reliable and effective.

The discriminant functions (DFs) and classification matrices (CMs) obtained by the standard, forward stepwise and backward stepwise modes of DA are shown in Tables 4 and 5. Both the standard mode and the forward stepwise mode were able to achieve discriminant accuracy rates of 80%, using 19 and 15 factors, respectively. However, the backward stepwise mode used only four factors (CODMn, Cu, As and Se) to achieve a discriminant accuracy rate close to 70%. The temporal DA showed that the four factors CODMn, Cu, As and Se were the most important parameters to distinguish the three periods obtained by clustering and accounted for most of the expected temporal changes in water quality.

The box plot of the four important parameters obtained by the backward discriminant analysis are shown in Fig 4. The average values of CODMn, Cu, As and Se showed the highest values in the first time period and showed a downward trend in the second and third time periods due to hydrological conditions (high flow period, flat flow period and low flow period), showing the characteristics of point pollution sources.

The results of the spatial analysis of DA were similar those of CA. The Wilks' lambda and chi-square values of each discriminant function were between 0.063 to 0.448 and 129.962 to 432.084 (p<0.01), respectively, indicating that the spatial discriminant analysis was credible and valid (Table 6).

The methods for obtaining the discriminant functions and classification matrices of the spatial DA were the same as those for the temporal DA and used the standard, forward stepwise, backward stepwise modes. The results are shown in Tables 7 and 8. The standard stepwise mode and the forward stepwise mode used 19 and 17 discriminant variables, respectively, and discriminant accuracy rates of 97.62% and 97.62% were obtained. However, in the

**Table 3. Results of temporal-DA for temporal variation.**

| Modes | DF | R | Wilks'lambda | Chi-square | P-level |
|---|---|---|---|---|---|
| Standard | 1 | 0.728 | 0.273 | 202.505 | 0.000 |
|  | 2 | 0.647 | 0.581 | 84.734 | 0.000 |
| Forward | 1 | 0.726 | 0.279 | 201.718 | 0.000 |
|  | 2 | 0.640 | 0.590 | 83.276 | 0.000 |
| Backward | 1 | 0.659 | 0.457 | 128.057 | 0.000 |
|  | 2 | 0.438 | 0.808 | 34.834 | 0.000 |

**Table 4. Classification functions coefficients for discriminant analysis of temporal variation.**

| Parameters | Standard mode | | | Forward stepwise mode | | | Backward stepwise mode | | |
|---|---|---|---|---|---|---|---|---|---|
| | 1st period | 2nd period | 3rd period | 1st period | 2nd period | 3rd period | 1st period | 2nd period | 3rd period |
| pH | 238.773 | 234.523 | 234.928 | 237.5 | 233.2 | 233.6 | | | |
| DO | 17.741 | 17.759 | 18.43 | 14.8 | 14.8 | 15.4 | | | |
| COD | 0.19 | 0.097 | 0.069 | 0.3 | 0.2 | 0.2 | | | |
| $NH_3$-N | 30.981 | 27.062 | 27.081 | 34.7 | 30.8 | 31 | | | |
| TP | 61.358 | 66.29 | 73.702 | 77.8 | 82.9 | 89.6 | | | |
| CODMn | -3.018 | -2.535 | -1.992 | -3.6 | -3.1 | -2.6 | 0.8 | 0.954 | 1.393 |
| F | 45.461 | 42.572 | 46.654 | 49.4 | 46.2 | 50.3 | | | |
| Oil | -56.343 | -62.48 | -46.918 | -51.2 | -57 | -41.5 | | | |
| $Cr^{6+}$ | 2.6 | 51.9 | -67.294 | 30.3 | 74.3 | -58.8 | | | |
| LAS | -193.536 | -183.126 | -202.094 | -239.9 | -230 | -249.4 | | | |
| Cu | 864.01 | 910.505 | 838.683 | 910.2 | 963.7 | 894.6 | 72.44 | 153.883 | 73.684 |
| Zn | 875.395 | 870.371 | 888.18 | | | | | | |
| Cd | 12491.85 | 23432.102 | 16533.812 | 17915.8 | 30535.2 | 22702 | | | |
| As | 9058.502 | 10366.496 | 8223.749 | 10494.3 | 11947.4 | 9881 | 1129.45 | 2514.299 | 640.057 |
| Hg | 35270.069 | 41507.369 | 42883.379 | | | | | | |
| CN | 799.035 | 850.65 | 870.161 | | | | | | |
| VP | -7102.505 | -7134.241 | -8590.354 | | | | | | |
| S | -4520.773 | -4189.78 | -4279.094 | -4408.3 | -4030.2 | -4189 | | | |
| Se | -11876.813 | -24283.73 | -17052.08 | -11393.9 | -24317.2 | -16832.4 | 11762.92 | 542.704 | 6614.182 |
| Constant | -1042.016 | -1008.92 | -1013.901 | -1007.4 | -973.8 | -977.2 | -12.74 | -11.188 | -9.39 |

backward step-by-step mode, the DA used only 7 discriminant parameters to produce a discriminant accuracy rate of 92.86%, which indicated that COD, TP, CODMn, F, LAS, Cu, and Cd were important parameters of the spatial variables.

Box and whisker plots of the discriminant parameters recognized by DA are given in Fig 5. In these seven groups of graphs, the minimum value of all factor averages is group C because group

**Table 5. Classification matrix for discriminant analysis of temporal variation.**

| Monitoring sites | Percent correct | Period assigned by DA | | |
|---|---|---|---|---|
| | | Group A | Group B | Group C |
| **Standard mode** | | | | |
| Group A | 78.57 | 22 | 2 | 6 |
| Group B | 80.36 | 0 | 45 | 11 |
| Group C | 78.57 | 6 | 45 | 66 |
| Total | 79.17 | 28 | 57 | 83 |
| **Forward stepwise mode** | | | | |
| Group A | 67.86 | 19 | 0 | 9 |
| Group B | 85.71 | 0 | 48 | 8 |
| Group C | 82.14 | 5 | 10 | 69 |
| Total | 80.95 | 24 | 58 | 86 |
| **Backward stepwise mode** | | | | |
| Group A | 39.29 | 11 | 1 | 16 |
| Group B | 71.43 | 0 | 40 | 16 |
| Group C | 76.19 | 8 | 12 | 64 |
| Total | 68.45 | 19 | 53 | 96 |

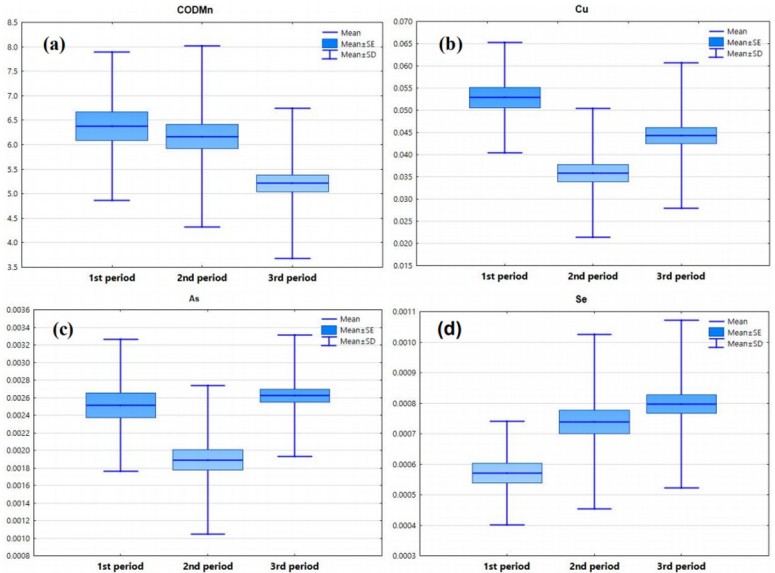

**Fig 4.** The (a)- (d) temporal variations: CODMn, Cu, As and Se.

C was a tributary of the Shuangji River, and the main water sources were from coal mines and reservoirs, which belonged to the clean water quality group. Among them, the maximum values of the average values of COD, TP, CODMn and LAS were all in group A because group A is the water quality of the upper Shuangji River, and the main sources of pollution in this area were the direct discharge of rural domestic sewage into the river and urban sewage plant drainage, which belonged to the group with poor water quality. The maximum value of the average value of F and COD was group B, and the main source of pollution was industrial wastewater.

## Principal component analysis/factor analysis

Before factor analysis, the Kaiser-Meyer-Olkin (KMO) and Bartlett's sphericity tests were performed to check the correlation and partial correlation between variables to judge whether the data were suitable for factor analysis [49]. The value of the KMO statistic ranges between 0 and 1 [50]. In the actual analysis, when the KMO statistic is above 0.7, the effect of the factor analysis of the data is considered to be better. The KMO result was 0.755, and Bartlett's sphericity result was 1342.53 ($p < 0.05$), showing that PCA can play an effective role in reducing dimensionality.

FA/PCA analysis is aimed at the standardized data and compares and analyses the composition patterns between water samples to determine the important factors that affect each water sample [51]. The PCA of all datasets yielded six (principal component)PCs, which explained

**Table 6. Results of spatial-DA for spatial variation.**

| Modes | DF | R | Wilk'slambda | Chi-square | P-level |
|---|---|---|---|---|---|
| Standard | 1 | 0.899 | 0.063 | 430.933 | 0.000 |
| | 2 | 0.819 | 0.328 | 173.689 | 0.000 |
| Forward | 1 | 0.897 | 0.064 | 432.084 | 0.000 |
| | 2 | 0.818 | 0.330 | 174.452 | 0.000 |
| Backward | 1 | 0.879 | 0.102 | 370.092 | 0.000 |
| | 2 | 0.743 | 0.448 | 129.962 | 0.000 |

**Table 7. Classification functions coefficients for discriminant analysis of spatial variation.**

| Parameters | Standard mode | | | Forward stepwise mode | | | Backward stepwise mode | | |
|---|---|---|---|---|---|---|---|---|---|
| | Group A | Group B | Group C | Group A | Group B | Group C | Group A | Group B | Group C |
| pH | 269.1 | 265.7 | 261 | 272.9 | 269.4 | 264.8 | | | |
| DO | 17 | 16.3 | 17.3 | | | | | | |
| COD | 1.1 | 1 | 0.8 | 1 | 0.9 | 0.8 | 0.21 | 0.21 | -0.01 |
| $NH_3$-N | 21.5 | 21.3 | 21 | | | | | | |
| TP | 130.1 | 156.1 | 120 | 134.6 | 161.6 | 123.5 | 35.14 | 68.34 | 28.34 |
| CODMn | 9.9 | 11.1 | 7.7 | 11 | 12.2 | 8.7 | 8.06 | 9.32 | 5.78 |
| F | 130.4 | 121.1 | 113.8 | 124.4 | 114.5 | 108.8 | 51.73 | 48.08 | 39.24 |
| Oil | -59.4 | -38.1 | -59.6 | -74.6 | -53.4 | -74.9 | | | |
| $Cr^{6+}$ | 1770.7 | 1633.7 | 1413.5 | 2126.6 | 1972.3 | 1776.1 | | | |
| LAS | -72.6 | -59.5 | -96.2 | -85.1 | -70.6 | -109.8 | 116.14 | 135.11 | 88.79 |
| Cu | 168.2 | 346 | 298.5 | 358.6 | 529.8 | 489.8 | -236.4 | -95.66 | -115.39 |
| Zn | 679.5 | 755.7 | 713 | 571.4 | 658.7 | 596.3 | | | |
| Cd | 142909.4 | 101402.6 | 123665.9 | 74515.5 | 36185.6 | 54165.7 | 86189.54 | 51500.86 | 63316.62 |
| As | 15541.5 | 14717 | 14327.2 | 12428.7 | 11962.3 | 10975.5 | | | |
| Hg | 149115.2 | 122729.6 | 138699.9 | 212009.9 | 181213.5 | 204055.1 | | | |
| CN | -1170.4 | -718.3 | -778 | -1835.2 | -1363.7 | -1448.2 | | | |
| VP | 17016 | 15247.1 | 11929.7 | 19340.7 | 17009.4 | 14689.8 | | | |
| S | -6953.9 | -6099.8 | -6384.7 | -7098.4 | -6186.5 | -6576.6 | | | |
| Se | -31329 | -28910.8 | -32671.3 | | | | | | |
| Constant | -1259.7 | -1248.3 | -1162.3 | -1194.4 | -1188 | -1095.2 | -71.5 | -89.71 | -39.86 |

68% of the total variance with eigenvalues > 1 (Table 1). The first PC (29.7% of the total variance) was correlated (loading >0.7) with COD, TP, Cu and VP. The third PC (9.2% of total variance) was correlated (loading>0.7) with LAS. However, the second, fourth, fifth and sixth PCs, although they accounted for the total variance of 10.4%, 7.2%, 5.9% and 5.5%, respectively, were not correlated (loading>0.7) with any of the parameters. Combining the local

**Table 8. Classification matrix for discriminant analysis of spatial variation.**

| Monitoring sites | Percent correct | Period assigned by DA | | |
|---|---|---|---|---|
| | | Group A | Group B | Group C |
| **Standard mode** | | | | |
| Group A | 98.80952 | 83 | 0 | 1 |
| Group B | 95.83334 | 1 | 23 | 0 |
| Group C | 96.66666 | 2 | 0 | 58 |
| Total | 97.61905 | 86 | 23 | 59 |
| **Forward stepwise mode** | | | | |
| Group A | 98.80952 | 83 | 0 | 1 |
| Group B | 95.83334 | 1 | 23 | 0 |
| Group C | 96.66666 | 2 | 0 | 58 |
| Total | 97.61905 | 86 | 23 | 59 |
| **Backward stepwise mode** | | | | |
| Group A | 92.85714 | 78 | 1 | 5 |
| Group B | 91.66666 | 2 | 22 | 0 |
| Group C | 93.33334 | 4 | 0 | 56 |
| Total | 92.85714 | 84 | 23 | 61 |

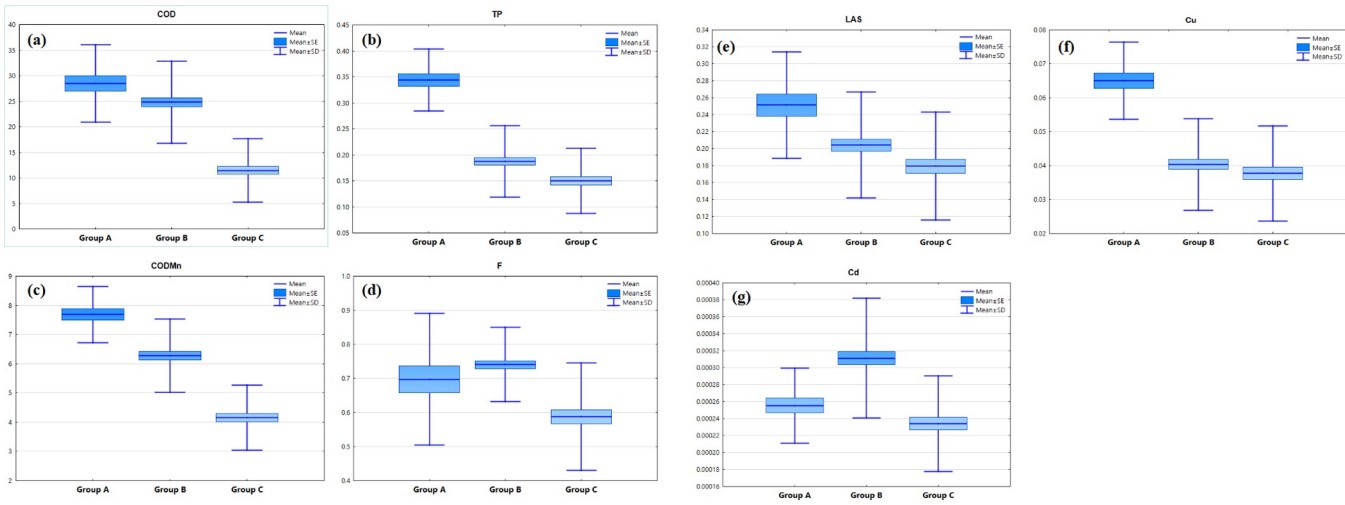

**Fig 5.** The (a)-(g) spatial variations: COD, TP, CODMn, F, LAS, Cu and Cd.

industrial structure and distribution, it characterizes emissions related to industrial industries such as handmade paper manufacturing, coking, chemical raw materials and chemical products manufacturing, and metal products, which are consistent with the current main industrial industries in Xinmi (Henan Province).

The Scree plot determines the number of PCs to keep by understanding the underlying data structure [52]. In this study, the Scree plot (Fig 6) showed a significant change in slope after the sixth eigenvalue. The original variable on the PC subspace is called the load, which was consistent with the correlation coefficient between the PC and the variable.

The axis of rotation defined by PCA will produce a new set of factors, each of which mainly involves a subset of the original variables, and the degree of overlap is as small as possible, so the original variables were divided into several independent groups [53]. Therefore, factor analysis (FA) of the current Shuangji River dataset further reduces the contribution of the

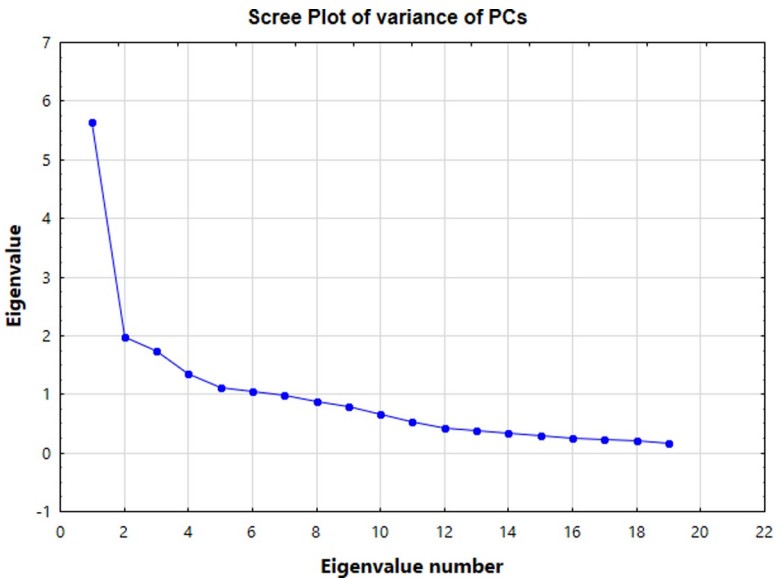

**Fig 6. Scree plot of variance of PCs.**

nonsignificant variables obtained from the PCA. The maximum variance rotation of the PC (original) explained the six different VFs with eigenvalues> 1, explaining approximately about 68% of the total variance. After the rotation by the maximum variance method, the value of PC was further revealed, and in VF, the participation of the original variable was clearer (Table 9). Liu et al. (2003) [54] classified the factor loadings as 'strong', 'moderate', and 'weak', which corresponded to absolute loading values of > 0.75, 0.75–0.50 and 0.50–0.30, respectively.

VF1 (17.7% of the total variance) had strong positive loadings on Cu and S and moderate positive loadings on $NH_3$-N and TP, indicating pollution from mineral composition and domestic sewage. This is because the main source of rivers in Xinmi City is drainage from domestic sewage plants, and the main source of pollution is domestic pollution sources, and the upstream of the tributaries are mainly coal and metal industries, causing some heavy metal pollution. VF2 (12.6% of total variance) had strong positive loadings on Cd and moderate positive loadings on Se and COD, indicating that the source was industrial wastewater pollution. This is related to the main paper industry, metal products, and steel casting manufacturing industries along the Shuangji River. VF3 (10.7% of total variance) had strong positive loadings on F and Oil and moderate positive loadings on $Cr^{6+}$, mainly manifested as fluoride pollution and heavy metal pollution. This clustering indicates that the source of pollution was the discharge of wastewater from the chemical industry, which is related to some chemical industries and metal manufacturing industries along the river. VF4, which explained 9.9% of the total variance, had moderate positive loadings on CODMn, $NH_3$-N and Zn, indicating that the pollution was from

**Table 9. Loadings of experimental variables (19) on significant principal components (with Varimax rotation) for the Shuangji River data set.**

| Parameters | VF1 | VF2 | VF3 | VF4 | VF5 | VF6 |
|---|---|---|---|---|---|---|
| pH | -0.16 | -0.012 | 0.025 | -0.149 | 0.69 | 0.025 |
| DO | -0.102 | -0.051 | 0.069 | -0.115 | -0.093 | **-0.827** |
| COD | 0.282 | 0.545 | 0.378 | 0.283 | -0.027 | 0.162 |
| $NH_3$-N | 0.527 | -0.062 | -0.346 | 0.524 | 0.195 | 0.036 |
| TP | 0.634 | 0.104 | 0.124 | 0.441 | -0.026 | 0.118 |
| CODMn | 0.244 | 0.257 | 0.304 | 0.647 | -0.043 | 0.109 |
| F | 0.03 | 0.236 | **0.794** | -0.097 | 0.022 | -0.311 |
| Oil | 0.192 | 0.028 | **0.718** | 0.151 | 0.248 | 0.199 |
| $Cr^{6+}$ | 0.133 | 0.345 | 0.516 | 0.373 | -0.246 | 0.225 |
| LAS | 0.446 | 0.067 | 0.195 | -0.295 | 0.631 | 0.127 |
| Cu | **0.809** | 0.191 | 0.177 | 0 | -0.156 | 0.043 |
| Zn | 0.011 | 0.036 | 0.012 | 0.648 | -0.229 | 0.057 |
| Cd | -0.026 | **0.771** | 0.148 | -0.077 | -0.202 | 0.309 |
| As | 0.308 | **0.74** | 0.152 | 0.105 | 0.068 | -0.056 |
| Hg | 0.574 | 0.214 | -0.039 | -0.044 | -0.483 | 0.124 |
| CN | 0.449 | 0.388 | 0.203 | -0.001 | -0.155 | 0.453 |
| VP | 0.537 | 0.18 | 0.286 | 0.209 | -0.189 | 0.416 |
| S | **0.828** | 0.067 | 0.057 | 0.153 | 0.162 | 0.04 |
| Se | 0.065 | 0.652 | -0.077 | 0.342 | 0.47 | -0.239 |
| Eigenvalue | 3.369 | 2.399 | 2.036 | 1.884 | 1.708 | 1.496 |
| %Total variance | 17.731 | 12.629 | 10.714 | 9.915 | 8.991 | 7.876 |
| Cumulative % variance | 17.731 | 30.36 | 41.074 | 50.989 | 59.98 | 67.856 |

Extraction Method: Principal Component Analysis.

Rotation Method: Varimax with Kaiser Normalization.

Rotation converged in 32 iterations.

mineral-related hydrochemistry and domestic sewage, mainly from the discharge of wastewater from paper-making enterprises, domestic sewage, and metal manufacturing wastewater into the river along the coast. VF5 (8.9% of total variance) had moderate positive loadings on pH and LAS, which can be interpreted as coming from detergents and personal necessities in domestic sewage. VF6, which explained 7.9% of the total variance, had moderate negative loadings on DO. This finding suggests that the pollutants in the water consumed a large amount of oxygen.

The FA/PCA results indicated that most changes were composed of soluble salts (natural) and organic pollutants (artificial). FA/PCA indicated that the main pollutants are COD, CODMn, $NH_3$-N, TP, Cu,$Cr^{6+}$, Zn, S, Se, Cd, F, Oil and LAS. These pollutants mainly come from domestic sewage discharge COD, CODMn, $NH_3$-N, TP; papermaking wastewater discharge COD, CODMn; textile industry, chemical product manufacturing wastewater discharge F, Oil and LAS; metal product manufacturing, optoelectronic device manufacturing and other industrial wastewater discharge Cu, $Cr^{6+}$, Zn, S, Se and Cd. The FA can identify the parameters that have the greatest contributions to changes in river water quality. The method of assessing the spatiotemporal changes in water quality based on FA/PCA have been applied to water quality evaluation at an early stage.

## Conclusions

Water quality monitoring programmes generate complex multi-dimensional data, which requires multivariate statistical processing to analyse and interpret its basic information. In this study, different multivariate statistical techniques were used to evaluate the spatial and temporal variations in the surface water quality of the Shuangji River. Cluster analysis (CA) divided the 12 months and 14 sampling points into three categories according to the similarity of river water quality characteristics and pollution. It provided an effective basis for the classification of surface water in the studied area and can effectively reduce the number of sampling points to analyse the river under the premise of lower loss of information. Discriminant analysis (DA) provides the best results for spatial and temporal analysis. It used only four factors (CODMn, Cu, As, Se) to distinguish the seasons temporally and achieved a 68% (79% reduction) accuracy rate and used only seven parameters (COD, TP, CODMn, F, LAS, Cu, and Cd) to allocate the three areas and achieve a 93% (63% reduction) accuracy rate. Although the FA/PCA pointed out the 7 parameters required to explain 68% of the data variability (37% of the original 19 parameters), only a small amount of data was reduced. However, the six VFs obtained from the PC indicated that the quality parameters of the river water were mainly divided into natural (soluble salts) and anthropogenic (organic pollution) components. Therefore, multivariate statistical techniques are an excellent exploration tool for analysing and interpreting complex datasets related to water quality and understanding their temporal and spatial changes.

## Supporting information

**S1 Raw data.**
(DOC)

## Acknowledgments

In this study, the authors are grateful for the data support provided by the Xinmi Branch of Zhengzhou Bureau of Ecology and Environment.

## Author Contributions

**Conceptualization:** Junzhao Liu.

**Data curation:** Dong Zhang, Hongbin Xu.

**Formal analysis:** Junzhao Liu.

**Funding acquisition:** Hongbin Xu.

**Investigation:** Junzhao Liu.

**Methodology:** Junzhao Liu.

**Project administration:** Hongbin Xu.

**Resources:** Dong Zhang, Hongbin Xu.

**Supervision:** Dong Zhang, Hongbin Xu.

**Writing – original draft:** Junzhao Liu.

**Writing – review & editing:** Dong Zhang, Qiuju Tang, Hongbin Xu, Shanheng Huang, Dan Shang, Ruxue Liu.

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
