## [Decision Letter · Decision Letter 0]

9 Oct 2020

PONE-D-20-20664

Water quality assessment and source identification of the Shuangji River (China) using multivariate statistical methods

PLOS ONE

Dear Dr. Zhang,

Thank you for submitting your manuscript to PLOS ONE. After careful consideration, we feel that it has merit but does not fully meet PLOS ONE’s publication criteria as it currently stands. Therefore, we invite you to submit a revised version of the manuscript that addresses the points raised during the review process.

We look forward to receiving your revised manuscript.

Kind regards,

Bing Xue, Ph.D.

Academic Editor

PLOS ONE

Journal Requirements:

2. In your Methods section, please provide additional location information of the study sites, including geographic coordinates for the data set if available.

3. In your Methods section, please provide additional information regarding the permits you obtained for the work. Please ensure you have included the full name of the authority that approved the study site access and, if no permits were required, a brief statement explaining why."

4. We note that Figure 1 in your submission contains map images which may be copyrighted.

We require you to either (a) present written permission from the copyright holder to publish this figure specifically under the CC BY 4.0 license, or (b) remove the figure from your submission:

b. If you are unable to obtain permission from the original copyright holder to publish this figure under the CC BY 4.0 license or if the copyright holder’s requirements are incompatible with the CC BY 4.0 license, please either i) remove the figure or ii) supply a replacement figure that complies with the CC BY 4.0 license. Please check copyright information on all replacement figures and update the figure caption with source information. If applicable, please specify in the figure caption text when a figure is similar but not identical to the original image and is therefore for illustrative purposes only.

6. Please include a caption for figure 4.

Reviewers' comments:

Reviewer's Responses to Questions

**Comments to the Author**

1. Is the manuscript technically sound, and do the data support the conclusions?

Reviewer #1: Partly

Reviewer #2: Partly

Reviewer #3: Yes

2. Has the statistical analysis been performed appropriately and rigorously? 

Reviewer #1: Yes

Reviewer #2: N/A

Reviewer #3: Yes

3. Have the authors made all data underlying the findings in their manuscript fully available?

Reviewer #1: No

Reviewer #2: Yes

Reviewer #3: Yes

4. Is the manuscript presented in an intelligible fashion and written in standard English?

Reviewer #1: No

Reviewer #2: No

Reviewer #3: Yes

5. Review Comments to the Author

Reviewer #1: The specific comments on the first review of your manuscript "Water quality assessment and source identification of the Shuangji River (China) using multivariate statistical methods" (PONE-D-20-20664) are as follows:

1) I highly recommend authors to improve the language quality.

2) Introduction should be re-written as it is hard to know the main insight of the paper. And key statistical methods are not well presented.

3) Line 78, The author should unify the accuracy of the values.

4) The format of the remarks in Table 1 needs to be unified with the format in the table.

5) The accuracy of the values in Table 3 should be unified.

6) Line 251-256 The Author need to add relevant literature as supplementary proof.

7) A table with higher pollution intensity needs to be added to as the support information .

8) Line 290, it needs better reasons to divide pollution levels.

9) The format of the ordinate of Figures 2, 4, 5 and 6 needs to be unified, and spaces need to be added before the brackets. The format is the new Roman.

10) Some minor typos and inappropriate use of words should be cleaned up.

11) Line 398-401, the expression is obscure.

12) The meaning of the topic needs to be further clarified and expanded, and the corresponding preface and discussion need to be further improved. Currently, the paper lacks discussion based on results (various correlations) and cannot be limited to the analysis of data values. It is necessary to use software such as spass to analyze the correlation of data. These aspects require the author to refer to the latest research progress at home and abroad, condense the scientific issues, and then start a systematic discussion around the basic results of this article. Secondly, the corresponding preface must be reorganized.

13) The reference section has a format error or lacks relevant information, such as: Reference 2, please carefully follow the submission requirements.

14) The authors only presented the results achieved, without really discussing them in a deeper discussion.

15) The relationship between major influencing factors and pollution sources is not given , and the universality of statistical methods is questionable.

Reviewer #2: A large number of observation and analysis has been done by the authors, and this manuscript has some extent scientific significance. However, However, the submission is under standard at its present stage. A major revision is recommened.

Some comments of this manuscript are outlined as follow.

1. What is the scientific significance of your study? Why readers should pay attention to your research?

2. Why the Shuangji River basin is chosen as the study area? Does it take more advantage over other regions? If yes, what these merits are?

3. Language needs further improvement.

Reviewer #3: Water quality assessment and source identification of the Shuangji River (China) using multivariate statistical methods

PONE-D-20-20664

Assessment of river water quality using multivariate statistical tool help to identify the sources and parameters that lead to water pollution and largely changes due to this pollution load.

The authors highlighted some physicochemical and trace metals that indicate pollution in the Shuangji river.

Through the authors claim this is the first study of water quality assessment of the Shuangji river, the methods of assessment are not novel.

Several studies (some of them cited by authors itself) used multivariate statistical techniques thus it lacks novelty.

The authors must clearly explain the novelty of his work.

Also, I do not find the research gap which motivates to conduct this study.

Specific comments:

Line 23-27- it is a well-established fact. Many of us did this already.

L 57 it is not a natural process, they are environmental factors

L 87 Study reported data for 2 yrs i.e. 2019 and 2020. How? 2020 is still going on. The authors also mentioned the Covid-19 problem in data collection. It looks the data is only for one year. Please correct.

L 95 mention N and E in geographical coordinates

L 463 – 465 again a well-known fact. Please provides some novel fining and scope of the study. What other researches should learn from your work. What is the future implementation of your work? Please make it more appealing.

Table 2 makes it more compact. Like in one row provide mean ± SD and min-max in the second row.

Fig. 6 transfer it in supplementary figures

The authors should provide details of the instruments used for analysis. Model and make of instruments. What is the analytical performance of the various instruments? Details of LOD, SRM is missing.

Provide the relevance of highly altered parameters with respect to human health, the scope of water treatment before discharge.

6. PLOS authors have the option to publish the peer review history of their article (what does this mean?). If published, this will include your full peer review and any attached files.

Reviewer #1: No

Reviewer #2: No

Reviewer #3: No

---

## [Author Response · Author response to Decision Letter 0]

26 Nov 2020

Dear Editors and Reviewers,

We highly appreciate the valuable comments of the editors and reviewers on our manuscript of PONE-D-20-20664'. The suggestions are quite helpful for us and we have incorporated them into the revised paper. We have checked the manuscript carefully, and revised it according to the comments of the editors and reviewers to improve its quality. As below, specific response to the comments were provided point by point. The revised sentences of the manuscript were highlighted in red in revised manuscript. We hope the editors and the reviewers will be satisfied with our responses to the comments and the revisions for the original manuscript.

Thank you very much.

Sincerely yours,

Dong Zhang

Reply to reviewer(s)' Comments:

Reviewer #1:

Comments:

The specific comments on the first review of your manuscript "Water quality assessment and source identification of the Shuangji River (China) using multivariate statistical methods" (PONE-D-20-20664) are as follows:

1. I highly recommend authors to improve the language quality.

Response: First of all, we would like to express our gratitude for your time-consuming review of our manuscript. According to your valuable comments, we have invited expert to revise our manuscript. We hope the reviewers and editors will be satisfied with the revisions for the original manuscript. Related revisions for the manuscript were made in red color according to actual circumstances and located in page 1 (lines 5 and 6), page 2 (lines 13 and 18), page 5 (lines 97-98 and 107), page 6 (lines 122 and 125), page 7 (lines 135, 140-141 and 147-152), page 8 (lines 153-158 and 161-167), page 9 (lines 175), page 10 (lines 185 and 200), page 12 (lines 201 and 205), page 2 (lines 222, 225, 227 and 229), page 14 (line 224 ), page 16 (lines 287 and 290), page 18 (lines 315, 323 and 325-326), page 19 (line 333), page 19 (line 336), page 21 (lines 344 and 349), page 22 (lines 352 and 359), page 23 (lines 363 and 373), page 24 (line 377), page 25 (lines 388-389 and 391-392), page 26 (lines 404, 408 and 415), page 27 (lines 417-420 and 425), page 28 (lines 428-429, 434, 438 and 440), page 29 (lines 458,462 and 465), page 30 (line 474).

2. Introduction should be re-written as it is hard to know the main insight of the paper. And key statistical methods are not well presented.

Response: Thank you very much for your valuable suggestion. Based on your comments, we have rewritten the introduction and added an introduction to statistical methods in the introduction. Meanwhile, the related description has been provided in the text in pages 3-5, lines 45-91 in red.

Introduction

Water is the material basis for the existence of earth creatures, and water resources are the primary condition for maintaining the sustainable development of the earth's ecological environment [1].With the increasing consumption of water resources, the contradiction between the supply and demand of water resources has intensified, which puts forward greater requirements for the utilization and protection of surface water resources [2].

The surface water quality of a region depends to a large extent on envirnmental factors (temperature changes, precipitation and soil erosion) and and human input (discharge of municipal and industrial wastewater and over-exploitation of water resources) [3]. Among them, the discharge of urban sewage and industrial wastewater is a continuous source of pollution, so effective control of sewage discharge is of great significance to the improvement of water quality [4, 5].

Surface water runoff is a seasonal phenomenon that is mainly affected by the climate of the basin [6]. In addition, seasonal changes in precipitation, surface runoff, interflow, groundwater flow, and pumping in and pumping out have a strong influence on the river flow and the subsequent pollutant concentration in the river [7].Therefore, correct identification of potential sources of surface water quality pollution is the basis and prerequisite for water quality management.

Shuangji River is a polluted river. Its main source of water comes from urban sewage treatment plants and paper-making sewage treatment plants. It not only plays an important role in assimilating or removing urban and industrial wastewater and farmland runoff, but is also the the main inland water resources used for household, industrial, and irrigation purposes [8], Therefore, it is necessary to prevent and control river pollution and have reliable water quality information for effective management. Given the spatial and temporal changes in river water chemistry, regular monitoring programmes are needed to reliably estimate water quality [9]. This leads large and complex data matrices composed of a large number of physical and chemical parameters, which are often difficult to interpret, making it challenging to draw meaningful conclusions [9].

Multivariate statistical analysis is a branch developed from classical statistics and is a comprehensive analysis method [11, 12]. It can analyze the statistical laws of multiple objects and multiple indicators when they are related to each other, including cluster analysis (CA) [13], discriminant analysis (DA) [14], principal component analysis (PCA) [15] and factor analysis (FA) [16]. Multivariate statistical analysis is a suitable tool for multi-component chemical and physical measurements for meaningful data reduction and interpretation [17]. It is a valuable tool for identifying factors and sources that may affect water systems and cause changes in water quality [18].

In this article, we took the Shuangji River as the research object for the first time, set up 14 main detection points along the river and detected and analyzed 19 physical and chemical parameters in water samples. The detection time lasted for 2 years.Different multivariate statistical techniques were used to analyse the obtained datasets, to analyse the similarity or dissimilarity between monitoring periods or monitoring points, to identify the water quality variables that cause the spatial and temporal changes of river water quality, and to determine the impact of water sources (natural and anthropogenic factors).

3. Line 78, The author should unify the accuracy of the values.

Response: Thank you very much for your valuable suggestion. According to your suggestions, we have checked 78 lines of sentences and confirmed their accuracy, In addition, we have carefully checked the whole manuscript to avoid similar mistakes elsewhere.

4. The format of the remarks in Table 1 needs to be unified with the format in the table.

Response: Thanks for the reviewer’s valuable advice. The problem pointed out by the reviewers has been checked according to your advice and the related information can be seen in page 9, line175.

Table 1 is as follows

Table 1. Water quality parameters associated with their acronyms, units and analytical methods used

S.N. Variables Acronyms Units Analytical methods

1 pH pH pH unit pH meter

2 Dissolved oxygen DO mg/L Prob metod

3 Chemical oxygen demand COD mg/L Dichromate method

4 Ammonical nitrogen NH3-N mg/L Spectrophotometric

5 Total phosphorus TP mg/L Ammonium molybdate Spectrophotometry

6 Chemical oxygen demand(Mn) CODMn mg/L Permanganate index method

7 Fluoride F mg/L Ion selective electrode

8 Petroleum oil Oil mg/L Infrared spectrophotometry

9 Linear alkylbenzene sulfonates LAS mg/L Methylene blue Spectrophotometry

10 Copper Cu mg/L FAAS

11 Zinc Zn mg/L FAAS

12 Cadmium Cd mg/L ETAAS

13 Arsenic As mg/L HGAAS

14 Mercury Hg mg/L CVAAS

15 Hexavalent chromium Cr6+ mg/L ETAAS

16 Total cyanide CN mg/L Pyridine barbituric acid Spectrophotometry

17 Volatile phenol VP mg/L Spectrophotometric Determination with 4-Amino-Antipyrin

18 Sulfide S mg/L Methylene blue Spectrophotometry

19 Selenium Se mg/L HGAAS

5. The accuracy of the values in Table 3 should be unified.

Response: Thanks for the reviewer’s valuable advice. The Table pointed out by the reviewers has been checked according to your advice and confirmed their accuracy, In addition, we have carefully checked the whole manuscript to avoid similar mistakes elsewhere.

Table 3 is as follows

Table 3. Results of temporal-DA for temporal variation

Modes DF R Wilks’lambda Chi-square P-level

Standard 1 0.728 0.273 202.505 0.000

 2 0.647 0.581 84.734 0.000

Forward 1 0.726 0.279 201.718 0.000

 2 0.640 0.590 83.276 0.000

Backward 1 0.659 0.457 128.057 0.000

 2 0.438 0.808 34.834 0.000

6. Line 251-256 The Author need to add relevant literature as supplementary proof.

Response: Thank you very much for your valuable suggestion. According to your valuable advice, we selected three references, related references were listed as follows. The related description has been provided in the text in pages 14-15, lines 253, 258 and 262, and References in red (in pages 36-37, lines 612-624).

[45] González MJG, Vallejo-Pascual M. The Application of Principal Component Analysis (PCA) for The Study of The Spanish Tourist Demand. Quaestiones geographicae. 2018 2018-01-01;37(4):43-52.

[46] Mohamad Asri MN, Mat Desa WNS, Ismail D. Combined Principal Component Analysis (PCA) and Hierarchical Cluster Analysis (HCA): an efficient chemometric approach in aged gel inks discrimination. AUST J FORENSIC SCI. 2020 2020-01-02;52(1):38-59.

[48] Ding S, Jia W, Su C, Zhang L, Liu L. Research of neural network algorithm based on factor analysis and cluster analysis. Neural Computing and Applications. 2011;20(2):297-302.

7. A table with higher pollution intensity needs to be added to as the support information.

Response: Thanks for the comment of the review. The issues pointed out by the reviewers has been added according to your advice. 

The table is as follows

Table 2-1 High pollution intensity in water bodies

Parameters M1 M2 Class III Class IV

pH Mean 7.56 7.63 6-9 6-9

DO (mg/L) Mean 7.57 8.56 5 3

COD (mg/L) Mean 30.08 26.92 20 30

NH3-N (mg/L) Mean 1.64 1.39 1.0 1.5

TP (mg/L) Mean 0.38 0.31 0.05 0.1

CODMn (mg/L) Mean 8.28 7.09 6 10

F (mg/L) Mean 0.55 0.84 1.0 1.5

Oil (mg/L) Mean 0.17 0.11 0.05 0.5

Cr6+ (mg/L) Mean 0.018 0.019 0.05 0.05

LAS (mg/L) Mean 0.27 0.24 

Cu (mg/L) Mean 0.06 0.07 1.0 1.0

Zn (mg/L) Mean 0.05 0.04 1.0 2.0

Cd (mg/L) Mean 0.00024 0.00027 0.05 0.1

As (mg/L) Mean 0.0029 0.0025 0.05 0.1

Hg (mg/L) Mean 0.00017 0.00015 0.0001 0.0005

CN (mg/L) Mean 0.015 0.013 0.2 0.2

VP (mg/L) Mean 0.0011 0.0010 0.005 0.1

S (mg/L) Mean 0.012 0.012 0.2 0.5

Se (mg/L) Mean 0.0009 0.0009 0.01 0.02

8. Line 290, it needs better reasons to divide pollution levels.

Response: Thank you very much for your careful review and the valuable suggestion. According to your opinion, we have added the reasons for the classification of pollution levels, and the related revision can be seen in page 17, lines 305-306.

9. The format of the ordinate of Figures 2, 4, 5 and 6 needs to be unified, and spaces need to be added before the brackets. The format is the new Roman.

Response: Thanks for the comment of the review. According to your valuable comments, we have checked the errors and revised them. Detailed information about the places where revisions was displayed in page 16, line 287, page 21, lines 343 and 344, page 24, lines 375-377, page 26, line 404. In addition, we have carefully checked the whole manuscript to avoid similar mistakes elsewhere (page 18, line 315)

The revised Fig. 2, Fig. 4, Fig. 5, Fig. 6 were shown as follows.

Fig 2. Dendrogram showing temporal clustering of monitoring periods

Fig 4. The (a)- (d) temporal variations: CODMn, Cu, As and Se

Fig 5.The (a)-(g) spatial variations: COD, TP, CODMn, F, LAS, Cu and Cd

Fig 6. Scree Plot of variance of PCs

The revised Fig 3 was shown as follows

Fig 3. Dendrogram showing spatial clustering of monitoring sites

10. Some minor typos and inappropriate use of words should be cleaned up.

Response: Thanks for the comment of the review. According to your valuable comments, we have checked the errors and revised them. Detailed information about the places where revisions were made in red color was displayed in page 1 (lines 5 and 6), page 2 (lines 13 and 18), page 5 (lines 97-98 and 107), page 6 (lines 122 and 125), page 7 (lines 135, 140-141 and 147-152), page 8 (lines 153-158 and 161-167), page 9 (lines 175), page 10 (lines 185 and 200), page 12 (lines 201 and 205), page 2 (lines 222, 225, 227 and 229), page 14 (line 224 ), page 16 (lines 287 and 290), page 18 (lines 315, 323 and 325-326), page 19 (line 333), page 19 (line 336), page 21 (lines 344 and 349), page 22 (lines 352 and 359), page 23 (lines 363 and 373), page 24 (line 377), page 25 (lines 388-389 and 391-392), page 26 (lines 404, 408 and 415), page 27 (lines 417-420 and 425), page 28 (lines 428-429, 434, 438 and 440), page 29 (lines 458,462 and 465), page 30 (line 474). Besides, we have invited expert to revise our manuscript. And some doctoral students in our research group have also tried our best to improve the language in the revised manuscript. We hope the reviewers and editors will be satisfied with the revisions for the original manuscript.

11. Line 398-401, the expression is obscure.

Response: Thank you very much for your valuable suggestion. We have checked this sentence carefully, it means that previous experimental studies have shown that when the characteristic value of the principal component is greater than 1, most of the factor variables can be expressed. It can be seen from Figure 6 that in the sixth principal component, the eigenvalue is greater than 1, that is, 19 factor variables can be expressed.

12. The meaning of the topic needs to be further clarified and expanded, and the corresponding preface and discussion need to be further improved. Currently, the paper lacks discussion based on results (various correlations) and cannot be limited to the analysis of data values. It is necessary to use software such as spass to analyze the correlation of data. These aspects require the author to refer to the latest research progress at home and abroad, condense the scientific issues, and then start a systematic discussion around the basic results of this article. Secondly, the corresponding preface must be reorganized.

Response: Many thanks for the reviewer’s affirmation to our manuscript. According to your suggestion, we have increased the discussion of the results. Detailed information about the places where revisions were made in red color was displayed in page 16, lines 278-286, page 25, lines 394-398, page 27, lines 421-424, page 28, lines 426-437, pages 28 and 29, lines 443-449. The introduction has also been revised, and the relevant revisions are in pages 3, 4 and 5, lines 45-91.

Introduction

Water is the material basis for the existence of earth creatures, and water resources are the primary condition for maintaining the sustainable development of the earth's ecological environment [1].With the increasing consumption of water resources, the contradiction between the supply and demand of water resources has intensified, which puts forward greater requirements for the utilization and protection of surface water resources [2].

The surface water quality of a region depends to a large extent on envirnmental factors (temperature changes, precipitation and soil erosion) and and human input (discharge of municipal and industrial wastewater and over-exploitation of water resources) [3]. Among them, the discharge of urban sewage and industrial wastewater is a continuous source of pollution, so effective control of sewage discharge is of great significance to the improvement of water quality [4, 5].

Surface water runoff is a seasonal phenomenon that is mainly affected by the climate of the basin [6]. In addition, seasonal changes in precipitation, surface runoff, interflow, groundwater flow, and pumping in and pumping out have a strong influence on the river flow and the subsequent pollutant concentration in the river [7].Therefore, correct identification of potential sources of surface water quality pollution is the basis and prerequisite for water quality management.

Shuangji River is a polluted river. Its main source of water comes from urban sewage treatment plants and paper-making sewage treatment plants. It not only plays an important role in assimilating or removing urban and industrial wastewater and farmland runoff, but is also the the main inland water resources used for household, industrial, and irrigation purposes [8], Therefore, it is necessary to prevent and control river pollution and have reliable water quality information for effective management. Given the spatial and temporal changes in river water chemistry, regular monitoring programmes are needed to reliably estimate water quality [9]. This leads large and complex data matrices composed of a large number of physical and chemical parameters, which are often difficult to interpret, making it challenging to draw meaningful conclusions [9].

Multivariate statistical analysis is a branch developed from classical statistics and is a comprehensive analysis method [11, 12]. It can analyze the statistical laws of multiple objects and multiple indicators when they are related to each other, including cluster analysis (CA) [13], discriminant analysis (DA) [14], principal component analysis (PCA) [15] and factor analysis (FA) [16]. Multivariate statistical analysis is a suitable tool for multi-component chemical and physical measurements for meaningful data reduction and interpretation [17]. It is a valuable tool for identifying factors and sources that may affect water systems and cause changes in water quality [18].

In this article, we took the Shuangji River as the research object for the first time, set up 14 main detection points along the river and detected and analyzed 19 physical and chemical parameters in water samples. The detection time lasted for 2 years.Different multivariate statistical techniques were used to analyse the obtained datasets, to analyse the similarity or dissimilarity between monitoring periods or monitoring points, to identify the water quality variables that cause the spatial and temporal changes of river water quality, and to determine the impact of water sources (natural and anthropogenic factors).

13. The reference section has a format error or lacks relevant information, such as: Reference 2, please carefully follow the submission requirements.

Response: Thanks for the comment of the review. According to your valuable comments, we have checked the errors and revised them. Detailed information about the places where revisions were made in red color was displayed reference. In addition, we have carefully checked the whole manuscript to avoid similar mistakes elsewhere.

14. The authors only presented the results achieved, without really discussing them in a deeper discussion.

Response: Thanks for your valuable comments. According to your suggestion, we have increased the discussion of the results, and the relevant information was also provided in the revised manuscript (in page 16, lines 278-286, in page 25, lines 394-398, page 27, lines 421-424, page 28, lines 426-437, pages 28 and 29, lines 443-449).

Since the Shuangji River is mainly a polluted river, the main body of the river comes from the sewage treatment plant along the bank, and the change in water quality reflects the change in the treatment effect of the sewage treatment plant. In summer, the sewage treatment plant has a better treatment effect, and the summer rainfall is large, and the river flow is large, so the river water quality in summer is better and divided into one category. In winter, the sewage treatment plant has poor water quality due to temperature and operation, and the rainfall is small, and the river flow is small. Therefore, the river water quality in winter is poor and divided into one category.

FA/PCA analysis is aimed at the standardized data and compares and analyses the composition patterns between water samples to determine the important factors that affect each water sample [50]. The PCA of all datasets yielded six (principal component)PCs, which explained 68% of the total variance with eigenvalues > 1 (Table 1). The first PC (29.7% of the total variance) was correlated (loading >0.7) with COD, TP, Cu and VP. The third PC (9.2% of total variance) was correlated (loading>0.7) with LAS. However, the second, fourth, fifth and sixth PCs, although they accounted for the total variance of 10.4%, 7.2%, 5.9% and 5.5%, respectively, were not correlated (loading>0.7) with any of the parameters. Combining the local industrial structure and distribution, it characterizes emissions related to industrial industries such as handmade paper manufacturing, coking, chemical raw materials and chemical products manufacturing, and metal products, which are consistent with the current main industrial industries in Xinmi (Henan Province).

VF1 (17.7% of the total variance) had strong positive loadings on Cu and S and moderate positive loadings on NH3-N and TP, indicating pollution from mineral composition and domestic sewage. This is because the main source of rivers in Xinmi City is drainage from domestic sewage plants, and the main source of pollution is domestic pollution sources, and the upstream of the tributaries are mainly coal and metal industries, causing some heavy metal pollution. VF2 (12.6% of total variance) had strong positive loadings on Cd and moderate positive loadings on Se and COD, indicating that the source was industrial wastewater pollution. This is related to the main paper industry, metal products, and steel casting manufacturing industries along the Shuangji River. VF3 (10.7% of total variance) had strong positive loadings on F and Oil and moderate positive loadings on Cr6+, mainly manifested as fluoride pollution and heavy metal pollution. This clustering indicates that the source of pollution was the discharge of wastewater from the chemical industry, which is related to some chemical industries and metal manufacturing industries along the river. VF4, which explained 9.9% of the total variance, had moderate positive loadings on CODMn, NH3-N and Zn, indicating that the pollution was from mineral-related hydrochemistry and domestic sewage, mainly from the discharge of wastewater from paper-making enterprises, domestic sewage, and metal manufacturing wastewater into the river along the coast. VF5 (8.9% of total variance) had moderate positive loadings on pH and LAS, which can be interpreted as coming from detergents and personal necessities in domestic sewage. VF6, which explained 7.9% of the total variance, had moderate negative loadings on DO. This finding suggests that the pollutants in the water consumed a large amount of oxygen.

The FA/PCA results indicated that most changes were composed of soluble salts (natural) and organic pollutants (artificial). FA/PCA indicated that the main pollutants are COD, CODMn, NH3-N, TP, Cu,Cr6+, Zn, S, Se, Cd, F, Oil and LAS. These pollutants mainly come from domestic sewage discharge COD, CODMn, NH3-N, TP; papermaking wastewater discharge COD, CODMn; textile industry, chemical product manufacturing wastewater discharge F, Oil and LAS; metal product manufacturing, optoelectronic device manufacturing and other industrial wastewater discharge Cu, Cr6+, Zn, S, Se and Cd. The FA can identify the parameters that have the greatest contributions to changes in river water quality. The method of assessing the spatiotemporal changes in water quality based on FA/PCA have been applied to water quality evaluation at an early stage.

15. The relationship between major influencing factors and pollution sources is not given , and the universality of statistical methods is questionable.

Response: Thanks for the reviewer’s valuable advice. The problem pointed out by the reviewers has been added according to your advice and the related information can be seen in discussion (in page 16, lines 278-286, in page 25, lines 394-398, page 27, lines 421-424, page 28, lines 426-437, pages 28 and 29, lines 443-449).

Reviewer #2:

Comments:

A large number of observation and analysis has been done by the authors, and this manuscript has some extent scientific significance. However, However, the submission is under standard at its present stage. A major revision is recommened.

1. What is the scientific significance of your study? Why readers should pay attention to your research?

Response: Many thanks for the reviewer’s affirmation to our manuscript. The scientific significance of this paper is to provide a basis for water quality analysis of polluted rivers.This is the first time that statistical analysis has been used to systematically evaluate the water quality of the Shuangji River and analyze the sources of pollutants in the Shuangji River. Therefore, this article can increase readers' interest.

2. Why the Shuangji River basin is chosen as the study area? Does it take more advantage over other regions? If yes, what these merits are?

Response: Thank you very much for your careful review and the valuable suggestion. The Shuangji River is an important river in Xinmi City. It carries the drainage of important sewage treatment plants in Xinmi City and the drainage of coastal enterprises. Its water sources are very complex, including paper industry water, domestic water, chemical industry water, and coal mine enterprise water. The complex water sources lead to the complexity of the water quality of the Shuangji River. If the water quality of the Shuangji River can be analyzed using scientific methods to simplify the complex water quality, this will be an example of solving complicated river pollution through scientific means. To provide a reference for the governance of other complex rivers, but also to provide scientific and effective methods for the work of the local government.

3. Language needs further improvement.

Response: Thanks for the comment of the review. According to your valuable comments, we have checked the errors and revised them. Detailed information about the places where revisions were made in red color was displayed in page 1 (lines 5 and 6), page 2 (lines 13 and 18), page 5 (lines 97-98 and 107), page 6 (lines 122 and 125), page 7 (lines 135, 140-141 and 147-152), page 8 (lines 153-158 and 161-167), page 9 (lines 175), page 10 (lines 185 and 200), page 12 (lines 201 and 205), page 2 (lines 222, 225, 227 and 229), page 14 (line 224 ), page 16 (lines 287 and 290), page 18 (lines 315, 323 and 325-326), page 19 (line 333), page 19 (line 336), page 21 (lines 344 and 349), page 22 (lines 352 and 359), page 23 (lines 363 and 373), page 24 (line 377), page 25 (lines 388-389 and 391-392), page 26 (lines 404, 408 and 415), page 27 (lines 417-420 and 425), page 28 (lines 428-429, 434, 438 and 440), page 29 (lines 458,462 and 465), page 30 (line 474). Besides, we have invited expert to revise our manuscript. And some doctoral students in our research group have also tried our best to improve the language in the revised manuscript. We hope the reviewers and editors will be satisfied with the revisions for the original manuscript.

Reviewer #3:

Comments:

Water quality assessment and source identification of the Shuangji River (China) using multivariate statistical methods.

1. Assessment of river water quality using multivariate statistical tool help to identify the sources and parameters that lead to water pollution and largely changes due to this pollution load. The authors highlighted some physicochemical and trace metals that indicate pollution in the Shuangji river. Through the authors claim this is the first study of water quality assessment of the Shuangji river, the methods of assessment are not novel. Several studies (some of them cited by authors itself) used multivariate statistical techniques thus it lacks novelty. The authors must clearly explain the novelty of his work. Also, I do not find the research gap which motivates to conduct this study.

Response: First of all, we would like to express our gratitude for your time-consuming review of our manuscript.The scientific significance of this paper is to provide a basis for water quality analysis of polluted rivers.This is the first time that statistical analysis has been used to systematically evaluate the water quality of the Shuangji River and analyze the sources of pollutants in the Shuangji River. Therefore, this article can increase readers' interest.

2. Line 23-27- it is a well-established fact. Many of us did this already.

L 57 it is not a natural process, they are environmental factors

L 87 Study reported data for 2 yrs i.e. 2019 and 2020. How? 2020 is still going on. The authors also mentioned the Covid-19 problem in data collection. It looks the data is only for one year. Please correct.

L 95 mention N and E in geographical coordinates

L 463 – 465 again a well-known fact. Please provides some novel fining and scope of the study. What other researches should learn from your work. What is the future implementation of your work? Please make it more appealing.

Table 2 makes it more compact. Like in one row provide mean ± SD and min-max in the second row.

Fig. 6 transfer it in supplementary figures

Response: Thanks for the comment of the review. According to your valuable comments, we have checked the errors and revised them. Detailed information about the places where revisions were made in red color was displayed in page 3, lines 52 and 53, page 3, lines 52 and 53。

The data of the study is two years (2018-2020). We have revised the time of the article. The relevant information is in page 1, line 3.

The geographic information of the Xinmi city has also been revised (page 5, line 94).

The Shuangji River is an important river in Xinmi City. It carries the drainage of important sewage treatment plants in Xinmi City and the drainage of coastal enterprises. Its water sources are very complex, including paper industry water, domestic water, chemical industry water, and coal mine enterprise water. The complex water sources lead to the complexity of the water quality of the Shuangji River. If the water quality of the Shuangji River can be analyzed using scientific methods to simplify the complex water quality, this will be an example of solving complicated river pollution through scientific means. To provide a reference for the governance of other complex rivers, but also to provide scientific and effective methods for the work of the local government.

We tried to modify Table 2 according to your suggestion, but the width of the modified table exceeds the layout, so that the table is divided into two, which affects the reading, so we boldly keep the original format.

3. The authors should provide details of the instruments used for analysis. Model and make of instruments. What is the analytical performance of the various instruments? Details of LOD, SRM is missing. Provide the relevance of highly altered parameters with respect to human health, the scope of water treatment before discharge.

Response: Thanks for the comment of the review. Based on your opinion, we have provided detailed information on the equipment used for analysis. Detailed information about the places where revisions were made in red color was displayed in page 4, lines 140-141 and 147-152, page 5, lines 153-158.

The pH value and DO value of each water sample were determined on site using digital pH values (JY-PH6.0) and DO measuring instruments (YT-RJY). The COD was measured by the dichromate reflux method (DH310C1COD) [20], and NH3-N was measured with Nessler’s reagent (NH3N-1040) [21]. The TP was measured by ammonium molybdate spectrophotometry (HM-812) [22], and the CODMn was measured by the permanganate index method (Thermo Scientific 3131) [22]. Fluoride (F) was measured using an ion-selective electrode (BHF5300) [24], and the total cyanides were analysed using pyridine barbituric acid spectrophotometry (TCN-508) [25], while sulphide (S) was determined using methylene blue spectrophotometry (ST201A) [26]. Petroleum hydrocarbons (Oil) were measured using infrared spectrophotometry (GC1290) [27]. The linear alkylbenzene sulfonates (LAS) were measured using methylene blue spectrophotometry (UltiMate3000) [28], and volatile phenols (VP) were measured using spectrophotometric determination with 4-amino-antipyrin (BELL) [29].

---

## [Decision Letter · Decision Letter 1]

4 Jan 2021

Water quality assessment and source identification of the Shuangji River (China) using multivariate statistical methods

PONE-D-20-20664R1

Dear Dr. Zhang,

We’re pleased to inform you that your manuscript has been judged scientifically suitable for publication and will be formally accepted for publication once it meets all outstanding technical requirements.

Kind regards,

Bing Xue, Ph.D.

Academic Editor

PLOS ONE

Additional Editor Comments (optional):

Reviewers' comments:

Reviewer's Responses to Questions

**Comments to the Author**

1. If the authors have adequately addressed your comments raised in a previous round of review and you feel that this manuscript is now acceptable for publication, you may indicate that here to bypass the “Comments to the Author” section, enter your conflict of interest statement in the “Confidential to Editor” section, and submit your "Accept" recommendation.

Reviewer #2: All comments have been addressed

Reviewer #3: All comments have been addressed

2. Is the manuscript technically sound, and do the data support the conclusions?

Reviewer #2: Yes

Reviewer #3: Yes

3. Has the statistical analysis been performed appropriately and rigorously? 

Reviewer #2: Yes

Reviewer #3: Yes

4. Have the authors made all data underlying the findings in their manuscript fully available?

Reviewer #2: Yes

Reviewer #3: Yes

5. Is the manuscript presented in an intelligible fashion and written in standard English?

Reviewer #2: Yes

Reviewer #3: Yes

6. Review Comments to the Author

Reviewer #2: The paper is substantial revised, and all my concerns is addressed. The ms can be accepted for publication now.

Reviewer #3: The authors attamed all the comments. Quality of the figure can be enhanced and the table should be compact.

I recommend acceptance of this article.

7. PLOS authors have the option to publish the peer review history of their article (what does this mean?). If published, this will include your full peer review and any attached files.

Reviewer #2: **Yes: **Cheng-Bang An

Reviewer #3: No

---

## [Editor Report · Acceptance letter]

6 Jan 2021

PONE-D-20-20664R1 

Water quality assessment and source identification of the Shuangji River (China) using multivariate statistical methods 

Dear Dr. Zhang:

I'm pleased to inform you that your manuscript has been deemed suitable for publication in PLOS ONE. Congratulations! Your manuscript is now with our production department. 

Kind regards, 

on behalf of

Professor Bing Xue 

Academic Editor

PLOS ONE